# Scanning Miniaturized Magnetometer Based on Diamond Quantum Sensors and Its Potential Application for Hidden Target Detection

**DOI:** 10.3390/s25061866

**Published:** 2025-03-17

**Authors:** Wookyoung Choi, Chanhu Park, Dongkwon Lee, Jaebum Park, Myeongwon Lee, Hong-Yeol Kim, Keun-Young Lee, Sung-Dan Lee, Dongjae Jeon, Seong-Hyok Kim, Donghun Lee

**Affiliations:** 1Department of Physics, Korea University, Seoul 02841, Republic of Korea; 2AI Laboratory, Chief Technology Officer (CTO) Division, LG Electronics Inc., Seoul 07796, Republic of Korea

**Keywords:** diamond NV centers, miniaturized sensor, scanning magnetometer

## Abstract

We have developed a miniaturized magnetic sensor based on diamond nitrogen-vacancy (NV) centers, combined with a two-dimensional scanning setup that enables imaging magnetic samples with millimeter-scale resolution. Using the lock-in detection scheme, we tracked changes in the NV’s spin resonances induced by the magnetic field from target samples. As a proof-of-principle demonstration of magnetic imaging, we used a toy diorama with hidden magnets to simulate scenarios such as the remote detection of landmines on a battlefield or locating concealed objects at a construction site, focusing on image analysis rather than addressing sensitivity for practical applications. The obtained magnetic images reveal that they can be influenced and distorted by the choice of frequency point used in the lock-in detection, as well as the magnitude of the sample’s magnetic field. Through magnetic simulations, we found good agreement between the measured and simulated images. Additionally, we propose a method based on NV vector magnetometry to compensate for the non-zero tilt angles of a target, enabling the accurate localization of its position. This work introduces a novel imaging method using a scanning miniaturized magnetometer to detect hidden magnetic objects, with potential applications in military and industrial sectors.

## 1. Introduction

With numerous breakthroughs and research initiatives, we are on the verge of entering a new era of quantum technology [1]. Quantum technology employs quantum mechanical objects, such as qubits, to address real-world challenges through innovative and distinct methods incompatible with their classical counterparts. Quantum sensors, one of the principal areas in quantum technology, aim to identify practical applications with the expectation of surpassing conventional sensors in sensitivity and resolution [2,3]. Diverse quantum platforms have been investigated, including superconducting quantum interference devices (SQUIDs [4]), entangled or squeezed lights [5], atomic vapor cells [6], and solid-state defects [2,3]. Diamond nitrogen-vacancy (NV) centers are exemplary defect-based qubits that uniquely offer the simultaneous advantages of high magnetic field sensitivity and spatial resolution [7,8,9]. A highly sensitive miniaturized diamond magnetic sensor can be developed due to its atomic dimensions and capability to function at ambient room temperature, with prospective applications in military, industrial, and biomedical fields [10,11,12,13,14].

As depicted in Figure 1, the NV center provides magnetic imaging capabilities over a wide range of length scales, from nanometers to micrometers and millimeters. Figure 1a illustrates a single-spin scanning magnetometer that offers nanometer-scale spatial resolution, advantageous for investigating the microscopic characteristics of magnetic materials and current-transport devices [15,16]. For instance, two-dimensional ferromagnetic materials [17], multiferroics [18], graphene [9,19], and Josephson junction devices [20] have been investigated to elucidate the magnetic domains and current distributions. However, this method is constrained by a slow scanning process and a small scan size, typically in the micrometer range. On the other hand, wide-field diamond microscopy captures magnetic images through a camera rather than employing a scanning method [21,22]. Consequently, it provides expedited imaging, with an order of second, across extensive sample areas exceeding several hundred micrometers; however, the spatial resolution is diminished to the diffraction-limited optical resolution, approximately several hundred nanometers (Figure 1b). This technique has been employed to visualize larger samples, including biological cells [21,23], neurons [24], geological rocks [25], and integrated circuits boards [26].

Despite the extensive studies conducted on the nanometer and micrometer scales, magnetic imaging at millimeter or larger scales has yet to be demonstrated. Here, we present magnetic imaging across several tens of centimeters with millimeter resolution utilizing a compact miniaturized magnetic sensor based on ensemble NV centers, integrated with a two-dimensional motorized scanning apparatus. For a proof-of-principle demonstration of magnetic imaging, we scanned the sensor over various sizes of permanent magnets concealed within a home-made diorama. Two distinct types of magnetic sensors are employed in the experiments, contingent upon the light source utilized for the optical excitation of the NV centers, as follows: one incorporates a built-in light-emitting diode (LED), while the other utilizes a fiber-coupled external laser. The optimal DC field sensitivity of 406 ± 2
$\mathrm{nT}/\sqrt{\mathrm{Hz}}$ is attained by assessing the linear response of the NV’s photoluminescence (PL) signals in relation to the frequency alterations induced by the Zeeman shift in the presence of a DC magnetic field.

In the measurements, we monitor changes in the PL signals at a constant carrier frequency, $f_{c}$, near the NV electron spin resonance (ESR). When the magnetic field is sufficiently weak, the frequency shift falls within the ESR linewidth, resulting in a well-defined scanned magnetic image. When the frequency deviates beyond the linewidth due to a strong magnetic field, the PL response at $f_{c}$ is no longer linear, necessitating the consideration of the non-linear response as well. This leads to distortions in the magnetic image, necessitating more complex image analysis. We examine magnetic images under various external magnetic fields and find a good agreement between the measured images and micromagnetic simulations, including the non-linear response. This is significant for sensor applications where the specimen under examination displays a broad range of magnetic fields, complicating image reconstruction.

This paper presents details of the scanning miniaturized magnetometer and its sensing mechanism and image analysis methods across various field magnitudes. We find that image analysis is essential for a comprehensive understanding of magnetic images, and consequently, the target objects. Using a toy diorama with embedded magnets, we emulate situations, such as the remote detection of landmines on a battlefield or concealed objects in a construction site. Given that varying magnitudes of magnetic fields can coexist in practical applications, it is essential to formulate a comprehensive analytical method or to develop suitable sensing techniques based on the field magnitude. Our work illustrates the capabilities of miniaturized magnetic sensors for military and industrial applications that require sensitive detection to locate concealed objects and targets.

## 2. Materials and Methods

### 2.1. Quantum Sensing Principle of Diamond NV Centers

Figure 2 illustrates the working principle of magnetic sensing based on diamond NV centers. The NV center comprises a substitutional nitrogen defect and a neighboring carbon vacancy within a diamond lattice (Figure 2a). The energy levels of a negatively charged NV center (note that we will call it NV center for the rest of the manuscript) reside within the bandgap of the diamond, where the NV center at the ground energy level exhibits S = 1 spin triplet states of $m_{s}=0$ and $m_{s}=\pm 1$, which are separated by 2.87 GHz at room temperature as a result of the crystal field [7,27]. The degenerated $m_{s}=\pm 1$ spin states are further split by a DC magnetic field along the NV’s crystal axis, $B_{\mathrm{NV}}$, due to the Zeeman effect. The amount of splitting corresponds to 2$\gamma_{\mathrm{NV}}B_{\mathrm{NV}}$, where $\gamma_{\mathrm{NV}}$ is the NV’s gyromagnetic ratio, 2.8 MHz/G, and is employed to probe DC magnetic fields, yielding the optimal sensitivity of approximately 1 $\mu T/\sqrt{\mathrm{Hz}}$ for a single NV center [3]. The optical excitation and readout of the NV’s spin states enables optically detected magnetic resonance (ODMR), characterized by a decrease in the NV’s photoluminescence signals at the spin transition frequencies between $m_{s}=0$ and $m_{s}=-1$, as well as $m_{s}=0$ and $m_{s}=+1$ [3,27]. Figure 2c illustrates a schematic of the ODMR spectrum, wherein the splitting between the transitions is used to detect the magnetic field along the NV axis.

The ODMR spectrum of ensemble NV centers exhibits four distinct pairs of spin resonances, which correspond to the four possible NV configurations within the diamond crystal structure as follows: $\left[111],\;[\bar{1}11],\;[1\bar{1}1\right],$ and $\left[11\bar{1}\right]$. In the absence of a magnetic field or under a very weak magnetic field, the spin resonances overlap and become nearly indistinguishable. Ensemble NV centers are commonly utilized in compact magnetic sensors, which, in principle, can provide enhanced sensitivity by a factor of $\sqrt{N}$ for shot noise-limited measurements, with *N* denoting the number of NV centers [2,3]. However, in real experiments, differences in coherence properties between single and ensemble NV centers, as well as spurious coupling effects in miniaturized sensor designs, can affect the extent of sensitivity enhancement. Improving sensitivity and addressing the issues of miniaturization remain important subjects for future studies.

### 2.2. Sensor Structure

As seen in Figure 3, we constructed two types of compact magnetic sensors, primarily distinguished by their light sources as follows: an LED in Figure 3a and a fiber-coupled laser in Figure 3b. We used a commercial diamond plate from Element Six—single crystal type 1b with the dimensions of 3 mm × 3 mm × 0.3 mm, a <100> crystal direction, and a nitrogen concentration of <200 ppm. The diamond was irradiated with 1.8 × 10^15^ electrons at 2 MeV and subsequently annealed at 900 °C for 2 h, resulting in an NV concentration of *N*~$2\times 10^{14}$. The diamond plate was affixed to a double split-ring resonator (DSRR) fabricated on a printed circuit board (PCB). The DSRR serves as an efficient and uniform microwave source to facilitate spin transitions around 2.87 GHz, with its design adopted from Refs. [28,29]. The resonator’s return loss measurement, *S*_11_, depicted in Figure 4a, indicates a resonance at 2.87 GHz with a quality factor *Q* of 395 (note that the simulated *Q* value is 1025, obtained from the full-wave numerical simulations based on CST Microwave Studio [30]). In order to adjust the DSRR resonance to align with the NV resonance, we positioned and adhered an additional rectangular copper plate, ~60 μm in thickness, adjacent to the ring resonator [28]. This modification resulted in a decrease in the *Q* factor to 160 (Figure 4b).

We utilized a commercial LED (M530D3, Thorlabs Inc., Newton, NJ, USA) and an external fiber-coupled solid-state laser (MGL-F-532, CNIlaser, Changchun, China) that emit a continuous wave (CW) green laser (λ = 532 nm) to excite the NV centers. In this experiment, we used a maximum laser power of ~16 mW with the LED and ~24 mW with the laser. The green light, from either the LED or the laser, was collimated using a lens (ACL12708U, Thorlabs, Newton, NJ, USA) or a fiber collimator (*f* = 7.5 mm, CFC8-A, Thorlabs, Newton, NJ, USA), followed by a dichroic mirror (86335, Edmund Optics, Barrington, NJ, USA) that selectively reflects light at wavelengths from λ = 350 nm to 596 nm and transmits light at wavelengths from λ = 612 nm to 950 nm. The green light is further concentrated at the center of the diamond utilizing a gradient-index (GRIN, 64519, Edmund Optics, Barrington, NJ, USA) lens affixed to the top surface of the diamond plate. Subsequent to optical excitation, the NV centers emit photons ranging from λ = 630 to 800 nm, which are collected by the same GRIN lens. The GRIN lens is utilized for its compact size and high efficiency in focusing and collecting light. From the collimated laser waist and the focal length of the GRIN lens, we estimated the detection volume in the diamond to be ~0.36 ${\mathrm{mm}}^{3}$ and the effective number of NV centers to be $\sim 3\times 10^{13}$.

The emitted photons traverse the dichroic mirror and focus on the photodiode in a silicon photodetector (PD; PDAPC1, Thorlabs, Newton, NJ, USA) after a pair of lenses (36168 and 49321, Edmund Optics, Barrington, NJ, USA). An optical filter (84746, Edmund Optics, Barrington, NJ, USA) is positioned in front of the photodetector to eliminate residual reflected green light after the dichroic mirror is used. All optical and electrical components, including the LED, lenses, PD, and DSRR, are secured in place within a plastic housing produced by a 3D printer. The DSRR, LED, and PD are connected via cables to external electronics and power sources. The PD signal is fed into a lock-in amplifier (MFLI, Zurich Instruments, Zurich, Switzerland) and is demodulated at 5 MHz, which is concurrently employed to the DSRR to modulate the microwave fields. The lock-in technique is employed to enhance the signal-to-noise ratio (SNR) of the PL signal.

### 2.3. Signal Processing and Sensitivity

Figure 5a,c illustrates an example of the ODMR spectrum and the lock-in result at zero applied magnetic field using the sensor in Figure 3a. The small splitting of 2.26 MHz in Figure 5a arises from the pre-existing non-zero strain parallel to the NV axis within the diamond crystal, which can vary from sample to sample [31,32]. Figure 5d illustrates the resonance height, referred to as the contrast (*C*), and the linewidth (Δ*f*) of the ODMR signal as a function of microwave power, *P*_MW_. Given an optical pumping power of 16 mW, both the ODMR contrast and linewidth increase with an increase in *P*_MW_. The overall behaviors of *C* and Δ*f* are well described in prior studies [33,34]. The inset in Figure 5c shows *C*/Δ*f*, which increases and saturates to 0.015 at *P*_MW_≈1 mW, and the minimum detectable signal, $B_{\min}$, is inversely proportional to *C*/Δ*f*.

Figure 5b shows the derivative of the ODMR data in Figure 5a with respect to frequency. The lock-in plot in Figure 5c is obtained by recording the demodulated PL signal in response to an oscillating microwave field at $f_{c}$ and repeating this process over the entire microwave frequency window by varying $f_{c}$. The lock-in data are essentially the same as those in Figure 5b, but we use the lock-in data for their enhanced SNR, as high-frequency noise is filtered out by the low-pass filter in the lock-in amplifier. In this study, we probed the local magnetic field and obtained magnetic images by analyzing the change in the lock-in signal while scanning the sensor over the target samples.

The Zeeman shift of the NV resonances can be written as follows:(1)$$\Delta f_{\pm }=2870\;\mathrm{MHz}\;\pm \;\sqrt{\delta^{2}+{\left|\Upsilon_{\mathrm{NV}}B_{\mathrm{NV}}\right|}^{2}}$$
where δ=2.26 MHz is the intrinsic splitting due to the crystal strain [31,32]. The sensitivity of the magnetic field, $\eta_{B}$, from the lock-in data can be calculated using the following equation:(2)$$\eta_{B}=\frac{\sqrt{\delta^{2}+{\left|\Upsilon_{\mathrm{NV}}B_{\mathrm{NV}}\right|}^{2}}}{\Upsilon_{\mathrm{NV}}^{2}\left|B_{\mathrm{NV}}\right|}\;\frac{\sigma }{\;{\left|dV/df\right|}_{\max}}\sqrt{\tau }$$
where ${\left|\mathrm{dV}/df\right|}_{\max}$ and σ represent the maximum slope and the standard deviation of the lock-in signal, respectively, and τ is the interrogation time, i.e., 1 s [34,35]. If $B_{\mathrm{NV}}\gg \;\delta$, Equation (2) can be approximated as follows:(3)$$\eta_{B}\;\approx \frac{1}{\Upsilon_{\mathrm{NV}}}\;\frac{\sigma }{\;{\left|dV/df\right|}_{\max}}\sqrt{\tau }$$
which assumes the linear dependence of the Zeeman on $B_{\mathrm{NV}}$. On the other hand, if $\delta \gg \;B_{\mathrm{NV}}$, Equation (2) can be approximated as follows:(4)$$\eta_{B}\;\approx \frac{1}{\Upsilon_{\mathrm{NV}}}(\frac{\delta }{\Upsilon_{\mathrm{NV}}\left|B_{\mathrm{NV}}\right|})\;\frac{\sigma }{{\left|dV/df\right|}_{\max}}\sqrt{\tau }$$
which introduces an additional factor of $\delta /\left(\Upsilon_{\mathrm{NV}}\left|B_{\mathrm{NV}}\right|\right)$, meaning the sensitivity also depends on $B_{\mathrm{NV}}$. Although we do not apply external magnetic fields to split the NV resonances beforehand in our current experiments, the linear regime of Equation (3) can be achieved either by applying an external magnetic field in advance or by ensuring the magnetic field to be probed is larger than the intrinsic transverse splitting.

To illustrate the sensitivity of the sensors used in the study, we calculate the sensitivity using Equation (3), assuming the linear dependence of the Zeeman shift on $B_{\mathrm{NV}}$. Figure 5e illustrates the calculated sensitivity from the lock-in data using Equation (3), showing the best sensitivity of $\eta_{B}=$ 628 ± 3 $nT/\sqrt{\mathrm{Hz}}$ at *P*_MW_≈1 mW. Note that the best sensitivity obtained from the other sensor in Figure 3b is $\eta_{B}=$ 406 ± 2 $nT/\sqrt{\mathrm{Hz}}$ at *P*_MW_≈1 mW. Although further improvements could be possible by increasing the microwave power, we observed a significant degradation in sensitivity when the microwave power exceeded ~8 mW, which we suspect arises from the generation of eddy current noise in the PD due to the intense microwave field [13]. This effect was observed only when DSRR was in close proximity to PD in the compact sensor configuration. To address this, a method is needed to filter out microwaves before they reach the photodetector, or alternatively, to separate them using fiber-coupled external photodetectors. These solutions will be explored in future studies.

### 2.4. Scanning Setup and Hidden Magnets Diorama

Figure 6a illustrates the scanning magnetometer setup with a diorama containing embedded magnets. The miniaturized magnetic sensors depicted in Figure 3 are mounted on a mobile stage capable of traversing a two-dimensional area utilizing two stepper motors and guiding rails. The scanning process involves initially advancing the stage along one axis, such as the *x*-axis, using a stepper motor, and subsequently maneuvering it along the *y*-axis with a second stepper motor. The total scan dimensions are 24.38 cm by 18.75 cm, with each step being 3.75 mm in size. A test magnetic sample utilized in this paper is a custom-built diorama with dimensions of 42 cm × 30 cm × 3 cm. We positioned the following three distinct sizes of permanent magnets as concealed targets: neodymium (Nd) disk magnets with (diameter, thickness) = (8 mm, 3 mm), (10 mm, 3 mm), and (8 mm, 12 mm). The diorama is located beneath the scanning stage, with its height along the *z*-axis regulated by a manual micrometer.

The diorama is intended to replicate a scenario in which landmines, explosive devices, or geological objects are buried underground, necessitating the use of unmanned magnetic sensors for remote and non-invasive detection [36,37]. Recent studies have shown the precise detection of hidden magnetic targets using miniaturized magnetic sensors, including fluxgate magnetometers [36,38] and optically pumped magnetometers (OPM), mounted on unmanned aerial vehicles (UAV) or drones [39,40]. To our knowledge, the NV-based magnetometer has yet to be used for this purpose; however, its compact size, high sensitivity, and ability for vector magnetometry indicate potential applications in military and industrial sectors.

Previous studies suggest that the magnetic fields from buried mines, measured a few meters above the ground, fall within the range from 1 nT to 100 nT [36,37,38,39,40]. Achieving this level of sensitivity would require improvements in several orders of magnitude over the current sensors presented in this paper. However, we anticipate that sensitivity can reach the necessary levels for practical applications, given the recent advancements in miniaturized diamond sensors [10,11,12,13,14]. In this study, we used a toy diorama sample to simulate the relevant battlefield scenarios, focusing on image analysis rather than addressing sensitivity for practical applications at this stage.

For magnetic imaging, we start scanning from the lower left corner of the diorama, i.e., (x, y) = (0 cm, 0 cm), and finish at the upper right corner of the diorama, i.e., (x, y) = (24.38 cm, 18.75 cm). For a single magnetic image, we execute a total of 1200 steps, comprising 40 steps along the *x*-axis and 30 steps along the *y*-axis, with an overall imaging duration of 6000 s. At each step, we allocate 5 s, comprising 1 s for measurement, 2 s for position adjustment, and 2 s for a pause prior to commencing measurement. The operation of stepper motors produces undesirable vibrations, necessitating a wait of approximately 2 s for the vibrations to dissipate before measurements can begin. We performed scanning measurements before and after the placement of the magnets on the diorama and subtracted the two images to eliminate background magnetic noises, including earth magnetic fields and those from the laboratory environment. We measured the lock-in signal at its peak and monitored its variation as the magnetic field from the sample changed. Figure 6b presents an example of the acquired diorama image alongside a simulated one. The distance between the NV centers, or the bottom surface of the sensor, and the top surface of the diorama is designated as *z*
≈ 10 cm. For the simulation, we utilize an open source software known as the Object-Oriented Micromagnetic Framework 1.2 (OOMMF [41]), with a mesh size of 0.1 mm × 0.1 mm × 0.1 mm. The obtained image distinctly illustrates the magnetic field profiles of the concealed magnets and is in good agreement with the simulation. Upon relocating the sensor nearer to the samples, however, we observed a discrepancy between the measured and simulated images, which will be discussed in detail in the following section.

## 3. Results and Discussion

### 3.1. Two Measurement Modes for Different DC Magnetic Field Magnitudes

In this study, we monitor the lock-in signal at a constant frequency, where changes in the signal result from the stray fields of the magnets. The fixed frequency measurement is commonly used in diamond magnetometry as it reduces total sensing time compared to the entire ESR spectrum across a wide frequency range [10,42,43,44,45]. In the context of magnetic imaging, however, the situation is more complex, necessitating the adjustment of the $f_{c}$ point based on the magnitude of the DC fields in relation to the ESR linewidth. Figure 7a illustrates an example of the measured lock-in signal, highlighting two specific frequency points. The frequency point indicated as ① is where the lock-in signal reaches its maximum. This frequency has been commonly employed due to the maximum slope of the original ODMR spectrum and the largest lock-in signal [42]. The second point, labeled as ②, represents the center frequency at which the lock-in signal varies linearly with the magnetic field (see the right panel in Figure 7a [10,43,45]). Although the lock-in signal may not be at its maximum, the linear response is advantageous for tracking small variations in the magnetic field.

In Figure 7, we compare the evolution of the lock-in signals based on the selection of $f_{c}$ points. Figure 7b shows the calculated lock-in signals as a function of the magnitude of the magnetic field along the NV axis, $B_{\mathrm{NV}}$. We modeled the ODMR spectrum as a linear combination of two Lorentzian functions, including intrinsic transverse splitting, and we obtained the lock-in signal, *L*_signal_, as the derivative of the ODMR spectrum with respect to frequency f, further normalized by its maximum value at f= ①, as follows:(5)$$L_{\mathrm{signal}}\left(f\right)=\left[\frac{{\left(\frac{1}{2}\Gamma_{1}\right)}^{2}\left(f-\Delta f_{-}\right)}{{\left\{{\left(f-\Delta f_{-}\right)}^{2}+{\left(\frac{1}{2}\Gamma_{1}\right)}^{2}\right\}}^{2}}+\frac{{\left(\frac{1}{2}\Gamma_{2}\right)}^{2}\left(f-\Delta f_{+}\right)}{{\left\{{\left(f-\Delta f_{+}\right)}^{2}+{\left(\frac{1}{2}\Gamma_{2}\right)}^{2}\right\}}^{2}}\right]$$
where $\Delta f_{\pm }=2870\;\mathrm{MHz}\;\pm \;\sqrt{\delta^{2}+{\left|\Upsilon_{\mathrm{NV}}B_{\mathrm{NV}}\right|}^{2}}$ [31,32]; δ=2.26 MHz represents the intrinsic splitting due to the crystal strain; and $\Gamma_{1}$ and $\Gamma_{2}$ represent the full width at half maximum (FWHM) of the ESR, respectively.

The calculated $L_{\mathrm{signal}}\left(f\right)$ at the two frequency points, $f=f_{c}=①$ and ②, are shown in Figure 7b. The plot is divided into the following two regions: I and II. The region I (where $B_{\mathrm{NV}}$
≈ 0–80 µT) exhibits the most pronounced and linear response in the lock-in signal when using the fixed carrier frequency point $f_{c}=$ ②, while the lock-in signal at $f_{c}=①$ shows minimal or nearly constant variations (see the right panel of Figure 7b). In contrast, region II (where $B_{\mathrm{NV}}$
≈ 80–150 µT) shows the most pronounced and linear variation in the lock-in signal when measured at $f_{c}=$ ①.

Figure 7b suggests that the optimal $f_{c}$ point should be adjusted according to the magnitude of $B_{\mathrm{NV}}$. When $B_{\mathrm{NV}}$ lies within region I, $f_{c}=$ ② is preferred, whereas $f_{c}=$ ① should be used when $B_{\mathrm{NV}}$ falls within region II. In practical situations, however, the magnetic field of the target can vary across the regions. Particularly, when using a compact magnetometer for imaging, the magnetic field may fluctuate significantly across the area surrounding the sample, potentially crossing between the regions. This variation can lead to complex images that complicate magnetic field analysis and, consequently, the determination of the target location.

### 3.2. Comparison of Magnetic Images Measured at the Frequency Points

To elucidate the effects more clearly, we conducted imaging experiments in which the magnetic field from samples is categorized into one of the two regions in Figure 7b. Figure 8 compares the mapping of changes in the lock-in signal, ${\Delta L}_{\mathrm{signal}}\left(f\right)$, measured at the frequency points ① and ②. We used two Nd disk magnets with (diameter, thickness) = (8 mm, 3 mm) in Figure 8a and (5 mm, 3 mm) in Figure 8b, where the magnetic field falls within region I. The data obtained at ② display the expected magnetic profiles, but the data obtained at ① show a reduced signal. For instance, the line-cut data in Figure 8a indicate ~77% of the lock-in signal for ① relative to ②. This discrepancy is more pronounced at lower magnetic fields, as illustrated in the line-cut data in Figure 8b, which shows only ~63% of the signal for ①. As expected from region I in Figure 7b, the discrepancy between the images obtained at ① and ② increases as the magnetic field from the sample becomes smaller.

Figure 7b and Figure 8 suggest that the magnetic field can be suppressed and the image distorted, depending on the $f_{c}$ point and the magnitude of the magnetic field. In order to understand the potential distortion in the images, we performed magnetic simulation based on our experimental conditions. The simulation procedures are illustrated in Figure 9. First, we modeled the magnet as a magnetic dipole whose magnetic moment, $\vec{m}$, is pre-assigned from the OOMMF simulation on the magnet as follows: (diameter, thickness) = (10 mm, 15 mm). Using Equation (6), we calculated the magnetic field profiles at z = 10 cm, as seen in Figure 9b [46].(6)$$\vec{B}\left(\vec{r}\right)=\frac{\mu_{0}}{4\pi }\left[\frac{3\vec{r}\left(\vec{m}\cdot \vec{r}\right)}{r^{5}}-\frac{\vec{m}}{r^{3}}\right]$$
where $\vec{B}\left(\vec{r}\right)$ is a vector magnetic field at a position $\vec{r}$, and $\mu_{0}$ represent the vacuum permeability. We then projected $\vec{B}\left(\vec{r}\right)$ along the NV axis using $B_{\mathrm{NV}}=\vec{B}\cdot \hat{u}\left(\theta ,\phi \right)$, where $\hat{u}\left(\theta ,\phi \right)=\left(\sin\theta \cos\phi \hat{x},\;\sin\theta \sin\phi \hat{y},\;\cos\theta \;\hat{z}\right)$, $\;\hat{x}$, $\;\hat{y}$, and $\;\hat{z}$ are the unit vectors, and θ and ϕ are the polar and azimuthal angles of the NV axis, as described in Figure 9a. Note that we only measure the absolute value of the magnetic field, i.e., $\left|B_{\mathrm{NV}}\right|$, due to the application of a zero magnetic field (not the field from the samples that we want to detect). Since there are four different NV crystal axes and their ESRs overlap at a non-zero or very weak magnetic field, we calculated the $\left|B_{\mathrm{NV}}\right|$ for all the NV axes, i.e., $\left(\theta ,\phi \right)=\left(54.7{}^{\circ},45{}^{\circ}\right)$, $\left(125.3{}^{\circ},135{}^{\circ}\right)$, $\left(125.3{}^{\circ},315{}^{\circ}\right)$, and $\left(54.7{}^{\circ},225{}^{\circ}\right)$ [31,32], and we repeated the process to obtain the combined magnetic images. As an example, the resulting magnetic image at ② is plotted in Figure 9c. Note that we identified the crystal axes beforehand, in a separate measurement, by applying an external magnetic field using a permanent magnet with known directions relative to the diamond crystal in our sensor configuration.

We also consider the tilt angles of the magnetometer relative to the magnet. Based on our experimental configuration, the magnet is tilted along the x- and y-axis by $\theta_{x}\approx 20{}^{\circ}$ and $\theta_{y}\approx 30{}^{\circ}$ relative to the diamond’s surface. We included the angles into the magnetic moment as ${\vec{m}}_{tilt}=R_{y}\left(\theta_{y}\right)\;R_{x}\left(\theta_{x}\right)\;\vec{m}$, where [31,32](7)$$R_{x}\left(\theta_{x}\right)=\left(\begin{array}{ccc} 1 & 0 & 0\\ 0 & cos\theta_{x} & -sin\theta_{x}\\ 0 & sin\theta_{x} & cos\theta_{x} \end{array}\right)\;\mathrm{and}\;R_{y}\left(\theta_{y}\right)=\left(\begin{array}{ccc} cos\theta_{y} & 0 & sin\theta_{y}\\ 0 & 1 & 0\\ -sin\theta_{y} & 0 & cos\theta_{y} \end{array}\right)$$

The simulated magnetic image, considering the tilted angles, $\left|B_{NV,\;tilted}\right|$, is shown in Figure 9d. Note that the nodal lines in Figure 9c,d correspond to cases where $\left|B_{NV}\right|=0$. As we measure the absolute magnitude of the magnetic field, abrupt changes in the images occur near $\left|B_{NV}\right|=0$. With the obtained $\left|B_{NV,\;tilted}\right|$, the amount of Zeeman shift is calculated using $\Delta f_{\pm }=2870\;\mathrm{MHz}\;\pm \;\sqrt{\delta^{2}+{\left|\Upsilon_{NV}B_{NV,\;tilted}\right|}^{2}}$ [31,32]. Afterward, the lock-in signal, $L_{signal}\left(f\right)$, in Equation (1) is calculated with the modified resonance frequencies at one of the fixed carrier frequencies $f_{c}=$ ① and ②. The final lock-in signal image at ② is plotted in Figure 9e. The magnet produces a wide range of magnetic fields, from 0 to ~150 μT, resulting in the lock-in signal evolving from a linear to a non-linear regime, as shown in Figure 7b. This highlights the importance of understanding the overall evolution of the lock-in data to analyze magnetic samples.

Figure 10 compares the measured and simulated lock-in images for magnets of various sizes as follows: (diameter, thickness) = (8 mm, 9 mm), (8 mm, 12 mm), (10 mm, 10 mm), and (10 mm, 15 mm). As the magnetic field increases from Figure 10a,d, the measured images exhibit more complex patterns, but these can be effectively identified by the simulation.

Moreover, identifying tilt angles from the magnetic images and compensating for them is crucial for accurately tracking the target’s location. For example, if one uses the maximum lock-in signal in Figure 10d to determine the magnet’s location, there can be errors of Δx ≈2.1 cm and Δy ≈−1.2 cm (assuming the original location is at the center of the simulated image), leading to the incorrect tracking of the target. We propose that vector magnetometry based on NV centers is a powerful method for identifying unknown tilt angles and significantly reducing the errors.

We performed additional simulations to identify the target location using vector magnetometry. Figure 11a shows an example of a magnetic field image at z = 10 cm, where the target (i.e., magnetic dipole) is tilted at angles of ($\theta_{x},\;\theta_{y})=\left(20{}^{\circ},\;30{}^{\circ}\right)$. We assume this image is a ’measured’ one. Note that, if the target location is determined by using the maximum magnetic field in this image, there are errors of Δx≈1.5 cm and Δy≈−0.6 cm relative to the correct location of the target (i.e., the center of the image). The goal of the simulation is to determine the tilt angles without prior knowledge and with minimum uncertainty.

From the image in Figure 11a, we first calculated the magnetic field components in a 3D space (i.e., $B_{x}$, $B_{y}$, and $B_{z}$) using vector magnetometry. We prepared four groups of NV centers (NV1, NV2, NV3, and NV4) that are well separated, as shown in Figure 2b. By measuring the Zeeman shifts in each NV group, we can extract $B_{x}$, $B_{y}$, and $B_{z}$ using the following relationships:(8)$$B_{x}=\frac{\sqrt{3}}{2}\left(B_{NV1}+B_{NV3}\right),\;B_{y}=\frac{\sqrt{3}}{2}\left(B_{NV1}-B_{NV2}\right),\;\mathrm{and}\;B_{z}=\frac{\sqrt{3}}{2}\left(B_{NV1}-B_{NV4}\right)$$

We then vary the angles, ($\theta_{x},\;\theta_{y})$, calculate the corresponding magnetic field components, and compare them with the ‘measured’ values from Equation (8) until the difference between them is minimized. The difference is defined as the averaged residue of each magnetic field components, calculated as follows:(9)$$Residue_{x}=\frac{\sum_{\vec{r}}\left(B_{x}\left(\theta_{x},\theta_{y}\right)-B_{x,NV}\right)}{Total\;data\;points\;in\;the\;2D\;image}$$(10)$$Residue_{y}=\frac{\sum_{\vec{r}}\left(B_{y}\left(\theta_{x},\theta_{y}\right)-B_{y,NV}\right)}{Total\;data\;points\;in\;the\;2D\;image}$$(11)$$Residue_{x}=\frac{\sum_{\vec{r}}\left(B_{z}\left(\theta_{x},\theta_{y}\right)-B_{z,NV}\right)}{Total\;data\;points\;in\;the\;2D\;image}$$

The total residue is then calculated as follows:(12)$$Total\;residue=Residue_{x}+Residue_{y}+Residue_{z}$$

As seen in Figure 11b, from the minimum of the ‘Total residue’, we obtained $\left(\theta_{x},\theta_{y}\right)=\left(19.80{}^{\circ}\;\pm \;0.45{}^{\circ},\;29.98{}^{\circ}\;\pm \;0.22{}^{\circ}\right)$. For the error analysis, we consider the sensitivity limit of the current sensor in the linear regime, i.e., $\eta_{B}=$ 406 $\mathrm{nT}/\sqrt{\mathrm{Hz}}nT/\sqrt{Hz}$ at P_MW_ ≈ 1 mW, as well as the standard deviation in the residue calculations from Equations (9)–(12). Finally, using the angles $\left(\theta_{x},\theta_{y}\right)=\left(19.80{}^{\circ},\;29.98{}^{\circ}\right)$, we corrected the ‘measured’ image in Figure 11a, as shown in Figure 11c. After correction, the errors in determining the target location z = 10 cm reduce to Δx≈0.05 mm and Δy≈−0.08 mm.

## 4. Conclusions

In this paper, we employed miniaturized NV magnetic sensors combined with a two-dimensional scanning stage to image concealed magnets in a toy diorama. To expediate the overall measurement process, we utilized the lock-in detection method for the NV’s ODMRs. We observed that the magnetic images could be suppressed and distorted depending on the choice of fixed carrier frequency points and the magnitude of the sample’s magnetic field. We performed magnetic simulations to analyze the evolution of the lock-in data as a function of the magnetic field and found good agreement between the measurements and simulations. A prior understanding of this evolution and subsequent image analysis is essential when using the miniaturized sensor to detect targets and extract their magnetic properties from the images. We also propose a method that uses vector magnetometry to compensate for the tilt angles of the target, enabling the accurate localization of its position. Our work presents a novel approach for using scanning miniaturized magnetometers and provides detailed analysis methods. To be applicable in real-world military and industrial environments, further improvements are necessary, including enhancing sensitivity to the order of $\mathrm{nT}/\sqrt{\mathrm{Hz}}nT/\sqrt{Hz}$ and incorporating vector magnetometry. Recent advances in miniaturized diamond sensors and sensing methodologies suggest that meeting these requirements simultaneously is feasible [46].

## Figures and Tables

**Figure 1 sensors-25-01866-f001:**
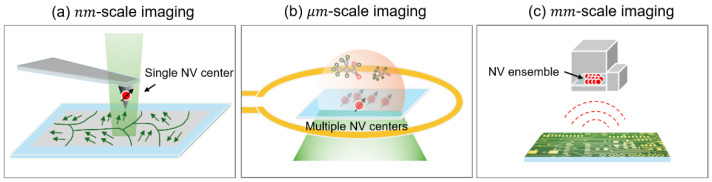
Schematics of (**a**) single-spin scanning magnetometry, (**b**) wide-field diamond microscopy, and (**c**) scanning miniaturized magnetometry.

**Figure 2 sensors-25-01866-f002:**
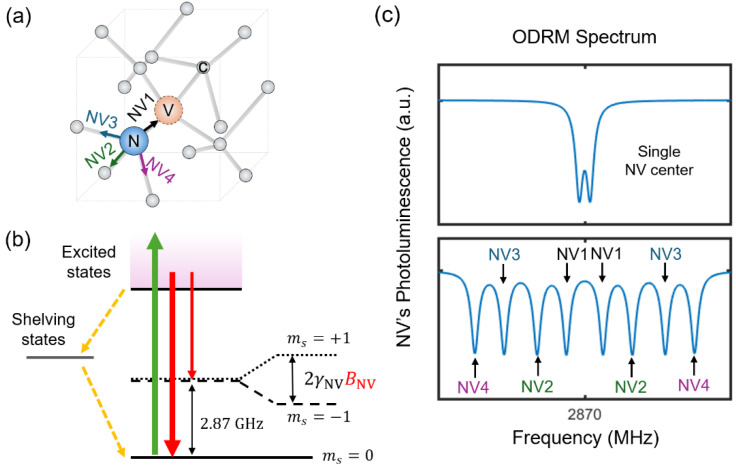
(**a**) Schematic of an NV center within a diamond lattice, showing four possible crystal axes for the NV centers. (**b**) Diagram illustrating the NV’s ground-state energy levels. The spin states of $m_{s}=0$ and $m_{s}=\pm 1$ are separated by 2.87 GHz at room temperature and the degenerated $m_{s}=\pm 1$ states can be split by an external magnetic field along the NV’s axis, $B_{\mathrm{NV}}$. Due to the intersystem transition via shelving states (dashed arrows), the NV’s photoluminescence signal is smaller for $m_{s}=\pm 1$ compared to $m_{s}=0$, as indicated by the relative widths of the red arrows. (**c**) Schematics of the ODMR spectra for a single NV center (upper plot) and ensemble NV centers (lower plot). In the ensemble spectrum, four pairs of spin resonances appear when exposed to an external magnetic field.

**Figure 3 sensors-25-01866-f003:**
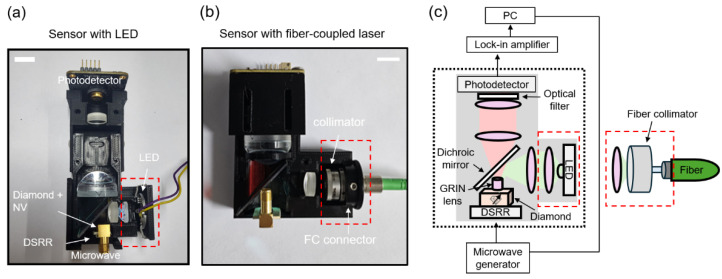
Pictures of miniaturized sensors based on two light sources: (**a**) an LED and (**b**) a fiber-coupled external laser. A DSRR is employed to efficiently deliver microwave fields from an external microwave generator to the NV centers. (**c**) Schematic of the sensor setups. Either the LED or the fiber-coupled laser provide excitation light at λ = 532 nm. The NV’s photoluminescence is collected by a photodetector after a GRIN lens, a dichroic mirror, pairs of lenses, and an optical filter. The lock-in detection technique is used to enhance the SNR of the photoluminescence signal.

**Figure 4 sensors-25-01866-f004:**
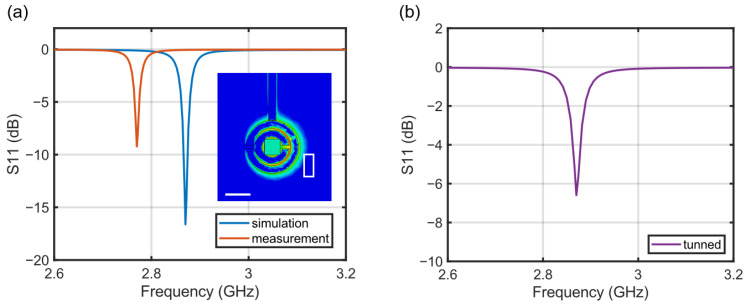
(**a**) DSRR return loss, *S*_11_, simulation (blue solid line) and measurement (orange solid line). The inset shows the simulated normal component of the microwave field around 2.87 GHz without the copper plate. The measured quality factor *Q* is 395, and the measured frequency deviates from the expected value from the simulation. To compensate for this difference, we positioned the copper plate near the diamond, as shown by the white box in the inset. Scale bar = 5 mm. (**b**) Measured *S*_11_ after matching the resonance frequencies between the NV center and the microwave resonator using the copper plate. The quality factor *Q* is reduced to 160 due to the insertion of the copper.

**Figure 5 sensors-25-01866-f005:**
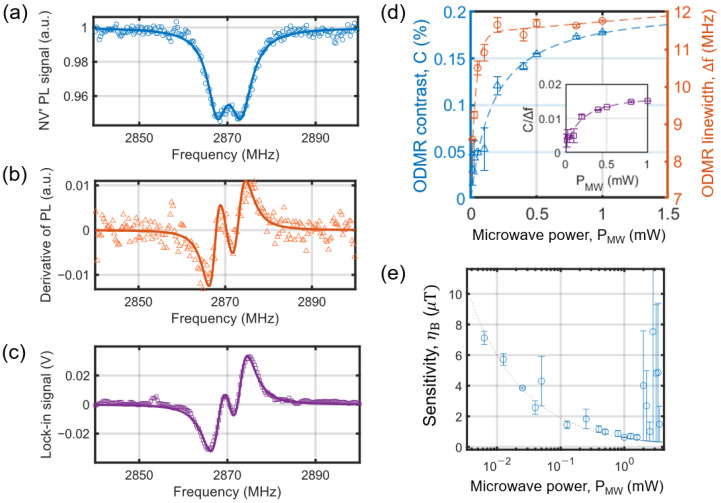
(**a**) ODMR spectrum measured by the sensor in Figure 3a. (**b**) Derivative of the ODMR data in (**a**). (**c**) Lock-in result of the same measurement in (**a**). (**d**) Contrast, *C*, and linewidth, Δ*f*, of the ODMR as a function of the microwave power, *P*_MW_. The inset plots *C*/Δ*f* as a function of *P*_MW_. (**e**) Calculated sensitivity, $\eta_{B}$, from the lock-in data using Equation (3) as a function of *P*_MW_.

**Figure 6 sensors-25-01866-f006:**
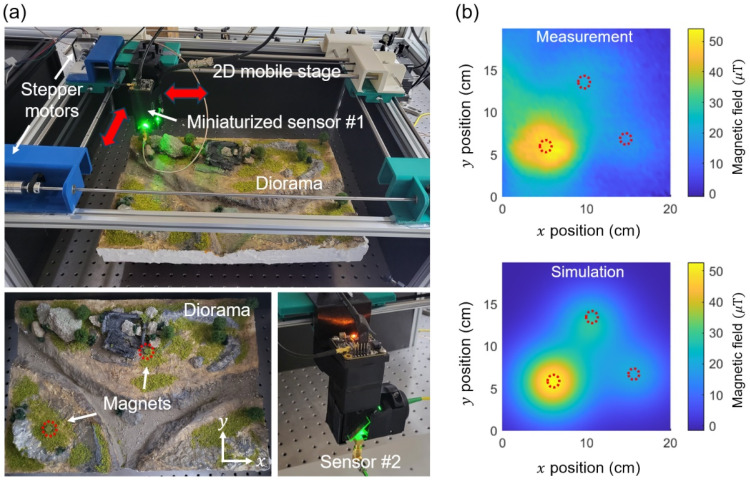
(**a**) Scanning miniaturized magnetometer setup. The miniaturized sensor, as shown in Figure 3a,b, is mounted on a 2D mobile stage that is controlled by two stepper motors. Diorama and concealed magnets are used to emulate landmines buried underground. The red arrows indicate the directions in which the sensor can be moved. (**b**) Measured and simulated magnetic field maps of three hidden Nd magnets. The magnetic image clearly reveals the locations of the hidden objects. The dotted circles in (**a**,**b**) indicate the positions of the concealed magnets.

**Figure 7 sensors-25-01866-f007:**
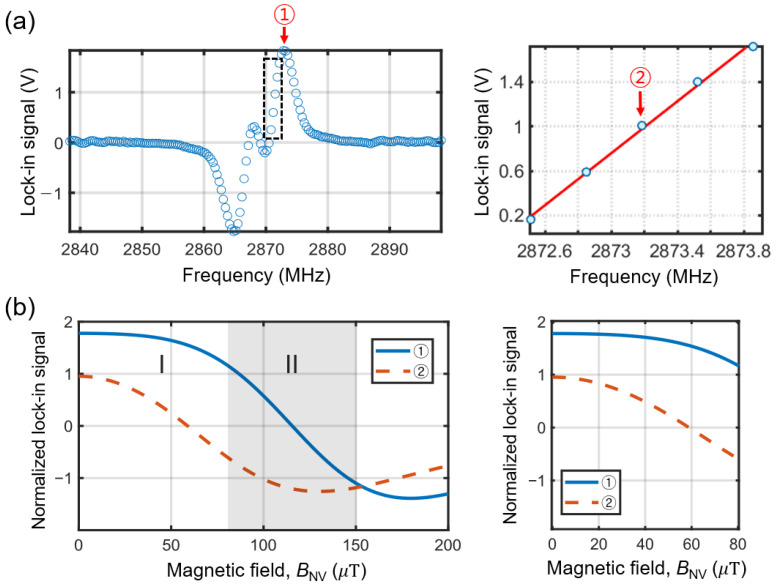
(**a**) Example of a lock-in measurement. The frequency point marked as ① corresponds to the point where the slope of the original ODMR spectrum and the lock-in signal are at their maximum. The right panel shows a zoomed-in view of the dashed rectangle in the left panel. The frequency point marked as ② represents the middle of the frequency range, where the lock-in signal varies linearly. (**b**) Normalized lock-in signal as a function of the magnetic field along the NV axis, $B_{\mathrm{NV}}$, calculated at $f_{c}$ = ① and ②. The right panel shows a zoomed-in view of the region I from the left panel.

**Figure 8 sensors-25-01866-f008:**
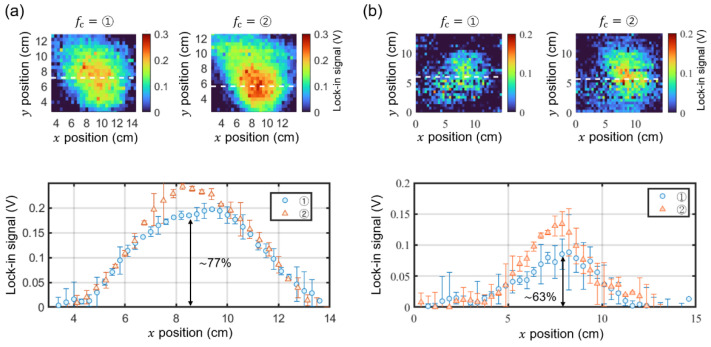
Magnetic images of Nd disk magnets with dimensions (diameter, thickness) = (8 mm, 3 mm) for (**a**) and (5 mm, 3 mm) for (**b**), measured at $f_{c}=$ ① and ②. The lower panels show the line-cut profiles along the dashed lines in the upper images. The lock-in signals measured at ① are ~ 77% (**a**) and ~63% (**b**) of the signals measured at ②.

**Figure 9 sensors-25-01866-f009:**
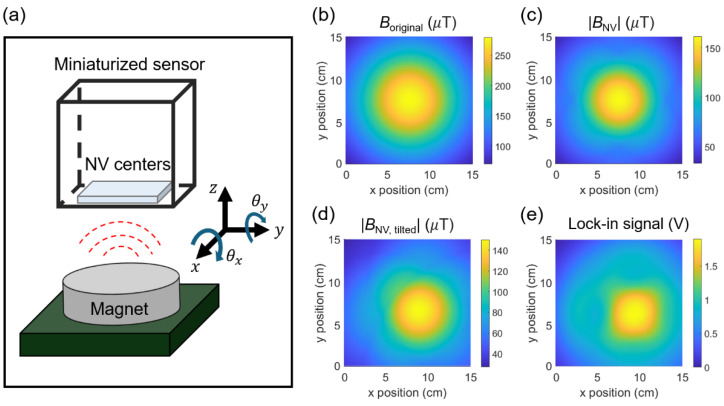
(**a**) Schematic of the relative angle between the NV centers in a miniaturized sensor and a magnet. $\theta_{x}\;$ and $\theta_{y}$ denote the tilt angles along the x- and y-axis. (**b**) Calculated magnetic field at z = 10 cm, $B_{original}$, obtained from the target. The dimensions of the magnet are (diameter, thickness) = (10 mm, 15 mm). (**c**) Absolute magnetic field projected onto the NV axis, $\left|B_{NV}\right|$. All NV axes are considered. (**d**) Magnetic image considering the relative tilt angles between the sensor and magnet, $\left|B_{NV,\;tilted}\right|$. The tilt angles used for the simulation are $\theta_{x}=20{}^{\circ}$ and $\theta_{y}=30{}^{\circ}$. (**e**) Calculated lock-in signal incorporating the Zeeman shift due to $\left|B_{NV,\;tilted}\right|$.

**Figure 10 sensors-25-01866-f010:**
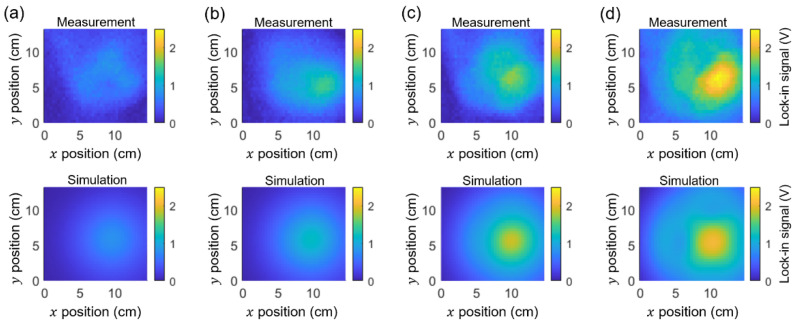
Measured and simulated lock-in images for magnets with dimensions (diameter, thickness) = (8 mm, 9 mm) (**a**), (8 mm, 12 mm) (**b**), (10 mm, 10 mm) (**c**), and (10 mm, 15 mm) (**d**). The measured images show stronger distortion, with more pronounced nodal lines. The simulations successfully capture the patterns observed in the measured images.

**Figure 11 sensors-25-01866-f011:**
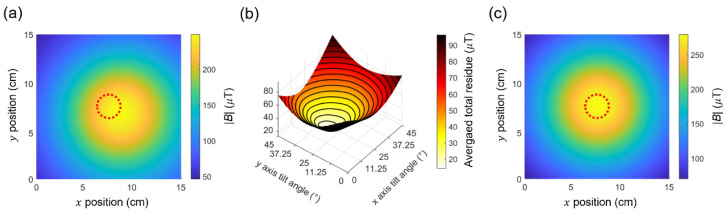
Correcting tilt angles using vector magnetometry (**a**) Simulated image of the absolute magnetic field, $\left|B\right|$, when the target is tilted at angles of ($\theta_{x},\;\theta_{y})=\left(20{}^{\circ},\;\;30{}^{\circ}\right)$. (**b**) Averaged total residue as a function of the tilt angles, $\left(\theta_{x},\;\theta_{y}\right)$. The minimum total residue occurs at $\left(\theta_{x},\theta_{y}\right)=\left(19.80{}^{\circ}\;,\;\;29.98{}^{\circ}\right)$. (**c**) Simulated image of the absolute magnetic field after correcting the tilt angles, as obtained from (**b**). The dotted circles in (**a**,**c**) denote the actual location of the target.

## Data Availability

The data used in this study are available upon request to the corresponding author.

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
