# Peer review of "Scanning Miniaturized Magnetometer Based on Diamond Quantum Sensors and Its Potential Application for Hidden Target Detection"

_sensors, 2025, doi:10.3390/s25061866_

Round 1

Reviewer 1 Report (Previous Reviewer 3)

Comments and Suggestions for Authors

The manuscript “Scanning Miniaturized Magnetometer Based On Diamond Quantum Sensors and Image Analysis for Target Localization” is devoted to a demonstration proof of the applicability of magnetic field sensors on NV centers for the tasks of magnetic scanning of surfaces. The latest version of the article (like the previous ones) is well structured and supplied with extensive illustrative material, as a result of which it is read with interest.

However, I should note that it still does not seem clear how the authors intend to scale the size of their search installation when moving from “a toy diorama” to real “battlefields”, especially taking into account the following features of the sensor they used:
i) sensitivity of 406 nT/Hz^(1/2) (L.69), given that, as the authors write, “the magnetic fields from buried mines, measured a few meters above the ground, fall within the range of 1 nT to 100 nT” (L. 272);
ii) lack of the ability to conduct vector measurements.

The authors state (L. 275) that “However, we anticipate that sensitivity can reach the necessary levels for practical applications, given recent advances in miniaturized diamond sensors [10-14]”. My question is: does any of the cited papers describe a vector (three-component) NV magnetometer with nanotesla resolution for DC signals that does not require millitesla bias fields? I am not aware of such articles, save for my be [Zheng, Huijie, et al. "Zero-field magnetometry based on nitrogen-vacancy ensembles in diamond." Physical Review Applied 11.6 (2019): 064068] – and this article proposes a method for measuring only one component of the field.

And once again I am forced to repeat: the lack of new scientific or engineering solutions still seems to me to be the main shortcoming of the manuscript.

I have a number of specific comments on the text of the article:

1. I find the new version of the title somewhat confusing.

2. L20 “This work introduces a novel imaging method using a scanning miniaturized magnetometer to detect hidden magnetic objects, with potential applications in military and industrial sectors”

– As already noted, I do not see anything new in this method. In my deep conviction, installing a magnetometer assembled according to a standard scheme on a 3D printer frame (or its analogue) is not a new solution, much less proof of the feasibility of using this magnetometer in real conditions.

3. L46 “However, this method is constrained by a slow scanning process and a small scan size, typically in the micrometer range”

– the method proposed by the authors is characterized (L61) by millimeter resolution, which is equally inappropriate in real conditions.

 4. L73 “In the measurements, we monitor changes in the PL signals at a constant frequency instead of sweeping across the entire NV electron spin resonance (ESR)”

 – the standard method used in magnetometry is not “sweeping across the entire NV electron spin resonance”, but locking the oscillator frequency to the resonance using frequency modulation and  feedback.

5. L199 “The small splitting of 2.26 MHz in Fig. 5(a) arises from pre-existing non-zero strain parallel to the NV axis within the diamond crystal”

– this value is not a constant, it varies widely from crystal to crystal.

6. L208 “The lock-in signal in Fig. 5(c) is essentially the same as the data in Fig. 5(b), but we use the lock-in data due to its enhanced SNR, as high-frequency noise is filtered out by a low-pass filter in the lock-in amplifier”

– it is unclear what “lock-in” the authors are talking about, if above (L73) they stated that the oscillator frequency is fixed.

7. L215 “where 2.26 MHz the intrinsic splitting due to the crystal strain”

– see remark 5.

8. L218 “represent the maximum slope and the standard deviation of the lock-in signal”

– what is the nature of the noise that determines the value of the standard deviation? The authors do not mention this.

9. L234 “Although further improvements could be possible by increasing microwave power, we observed a sharp increase in sensitivity when the microwave power exceeded ~ 8 mW, which we suspect arises from the generation of eddy current noise in the PD due to the intense microwave field

– does “increase in sensitivity” here mean deterioration or improvement?

10. L311 “The fixed frequency measurement is commonly used in diamond magnetometry as it reduces total sensing time compared to the entire ESR spectrum throughout an wide frequency range”

– see note 4.

11. L317 “The frequency point indicated as (1) is where the lock-in signal reaches its maximum. This frequency has been commonly employed due to the largest lock-in signal”

– This statement is absolutely incorrect, since this point is characterized by a zero derivative of the lock-in signal, and, accordingly, by the lack of sensitivity of this signal to changes in the magnetic field.

12. L348 “In contrast, the region II (80 − 150 µT) shows the most pronounced and linear variation of the lock-in signal when measured at (1)”

– This happens precisely because the frequency, which in the zero field corresponded to point (1), ends up in a linear section as the field increases.

13. L351 “Figure 7(b) suggests that the optimal fixed frequency point should be adjusted according to the magnitude of BNV

– this statement seems obvious, as well as all the results in this section.

14. L392 “Since there are four different NV crystal axes and their ESR resonances overlap at non-zero or very weak 393 magnetic field, we calculated BNV for all the NV axes”

– the authors do not mention anywhere how they measured the position of the crystal axes relative to the coordinate system used.

15. L433 “We propose that vector magnetometry based on NV centers is a powerful method for identifying unknown tilt angles and significantly reducing the errors”

– it is great that the authors fulfilled the reviewer’s request and added a numerical calculation, but they did not do the most important thing – namely, they did not show how a vector magnetometer scheme that works in the earth’s field can be implemented.

Overall, I don't think the article has been improved in any way by the revisions, and I still think it would be more suited to a popular journal rather than Sensors.

Author Response

Reviwer#1

Comments and Suggestions for Authors

The manuscript “Scanning Miniaturized Magnetometer Based On Diamond Quantum Sensors and Image Analysis for Target Localization” is devoted to a demonstration proof of the applicability of magnetic field sensors on NV centers for the tasks of magnetic scanning of surfaces. The latest version of the article (like the previous ones) is well structured and supplied with extensive illustrative material, as a result of which it is read with interest.

However, I should note that it still does not seem clear how the authors intend to scale the size of their search installation when moving from “a toy diorama” to real “battlefields”, especially taking into account the following features of the sensor they used:

  1. i) sensitivity of 406 nT/Hz^(1/2) (L.69), given that, as the authors write, “the magnetic fields from buried mines, measured a few meters above the ground, fall within the range of 1 nT to 100 nT” (L. 272);
  2. ii) lack of the ability to conduct vector measurements.

Authors: We thank the Reviewer for the helpful suggestions. However, as we have mentioned multiple times in previous responses, the manuscript focuses on addressing potential imaging challenges when applying miniaturized quantum sensors for two-dimensional imaging. To illustrate this, we used a toy diorama to highlight these issues and propose imaging analysis methods and possible solutions. We agree with the Reviewer that to apply the current setup to real battlefields applications, improvements are needed to achieve sensitivity in the 1 nT to 100 nT range and to perform vector magnetometry. However, these points are not the primary focus of the paper and are intended for future research. It seems the Reviewer is suggesting that the authors develop a better sensor and conduct outdoor measurements in real-world scenarios. Again, this manuscript primarily focuses on imaging analysis when using the sensor for scanning experiments. Note that the toy diorama serves as an example, but it does not limit the potential applications of the setup beyond the scenarios we have suggested.

The authors state (L. 275) that “However, we anticipate that sensitivity can reach the necessary levels for practical applications, given recent advances in miniaturized diamond sensors [10-14]”. My question is: does any of the cited papers describe a vector (three-component) NV magnetometer with nanotesla resolution for DC signals that does not require millitesla bias fields? I am not aware of such articles, save for my be [Zheng, Huijie, et al. "Zero-field magnetometry based on nitrogen-vacancy ensembles in diamond." Physical Review Applied 11.6 (2019): 064068] – and this article proposes a method for measuring only one component of the field.

Authors: As mentioned in the previous questions, the reviewer highlighted two key points regarding the application of the sensor in real-world battlefields: nanotesla sensitivity and vector magnetometry. We addressed the first point by stating the sentence (L275) with the relevant references (10-14) to illustrate that miniaturized diamond sensors can indeed achieve nanotesla sensitivity. Regarding the second point, many lab-scale experiments using ensemble NV centers have successfully demonstrated vector magnetometry, and these methodologies can be applied to miniaturized sensors. While satisfying both nanotesla sensitivity and vector magnetometry in miniaturized sensors has yet to be fully demonstrated, except in one recent study [Jonas Homrighausen et al., “Microscale fiber-integrated vector magnetometer with on-tip field biasing using N-V ensembles in diamond microcrystals,” Physical Review Applied 22, 034029 (2024)], we believe that continued development of our setup can meet these requirements.

We have added the following sentences in the conclusion and cited the article in the reference section to more clearly illustrate the potential perspectives of the sensor.

“Our work presents a novel approach for using scanning miniaturized magnetometers and provides detailed analysis methods. To be applicable in real-world military and industrial environments, further improvements are necessary, including enhancing sensitivity to the order of nT/√Hz and incorporating vector magnetometry. Recent advances in miniaturized diamond sensors and sensing methodologies suggest that meeting these requirements simultaneously is feasible[46].”

  1. Homrighausen, J.; Hoffmann, F.; Pogorzelski, J.; Glösekötter, P.; Gregor, M.; Microscale fiber-integrated vector magnetometer with on-tip field biasing using N-V ensembles in diamond microcrystals. Physical Review Applied 2024, 22, 034209, doi: 10.1103/PhysRevApplied.22.034029.

And once again I am forced to repeat: the lack of new scientific or engineering solutions still seems to me to be the main shortcoming of the manuscript.

Authors: As mentioned above, achieving both nanotesla sensitivity and vector magnetometry has only recently been demonstrated in a single article and remains one of key topics in most of the miniaturized sensor research. The reviewer has repeatedly argued that the absence of these capabilities makes our manuscript scientifically unimportant. However, we maintain that our work successfully demonstrates two-dimensional imaging with a miniaturized diamond sensor and provides detailed image analysis methods, which are also critical issues when applying the sensor to future real-world applications.

I have a number of specific comments on the text of the article:

  1. I find the new version of the title somewhat confusing.

Authors: We revised the original title based on the reviewer’s suggestion, changing it from “Millimeter-Resolution Scanning Miniaturized Magnetometer Based On Diamond Quantum Sensor” to “Scanning Miniaturized Magnetometer Based On Diamond Quantum Sensors and Image Analysis for Target Localization”

After further consideration, we have refined the title to “Scanning Miniaturized Magnetometer Based On Diamond Quantum Sensors And Its Potential Application For Hidden Target Detection”

  1. L20 “This work introduces a novel imaging method using a scanning miniaturized magnetometer to detect hidden magnetic objects, with potential applications in military and industrial sectors”

– As already noted, I do not see anything new in this method. In my deep conviction, installing a magnetometer assembled according to a standard scheme on a 3D printer frame (or its analogue) is not a new solution, much less proof of the feasibility of using this magnetometer in real conditions.

Authors: The reviewer has repeatedly argued that the absence of these capabilities makes our manuscript scientifically unimportant. However, we maintain that our work successfully demonstrates two-dimensional imaging with a miniaturized diamond sensor and provides detailed image analysis methods, which are also critical issues when applying the sensor to future real-world applications.

  1. L46 “However, this method is constrained by a slow scanning process and a small scan size, typically in the micrometer range”

– the method proposed by the authors is characterized (L61) by millimeter resolution, which is equally inappropriate in real conditions.

Authors: We used the toy diorama as an example to demonstrate the scanning miniaturized magnetometer, but it does not limit the potential applications of the setup beyond the scenarios we have suggested. For example, the sensor could be positioned near the surface of electronic devices to investigate their performance by imaging current profiles. In this case, spatial resolution can range from millimeters to centimeters. The sentence (L46) serves as a general description, highlighting the advantages of the method over other approaches, such as scanning magnetometry and wide-field microscopy.

  1. L73 “In the measurements, we monitor changes in the PL signals at a constant frequency instead of sweeping across the entire NV electron spin resonance (ESR)”

 – the standard method used in magnetometry is not “sweeping across the entire NV electron spin resonance”, but locking the oscillator frequency to the resonance using frequency modulation and  feedback.

Authors: In NV measurements, a full ESR spectrum provides a comprehensive understanding of the magnetic field, including both magnitude and direction, across a wide dynamic range. Of course, ESR at a fixed frequency is also commonly used and beneficial for certain applications, such as faster measurements within a narrow frequency window. Indeed, we employed the latter approach in our manuscript. The sentence (L73) is intended to help readers familiar with the diamond NV centers recognize that we used this method in our experiments.

  1. L199 “The small splitting of 2.26 MHz in Fig. 5(a) arises from pre-existing non-zero strain parallel to the NV axis within the diamond crystal”

– this value is not a constant, it varies widely from crystal to crystal.

Authors: We did not state that the intrinsic splitting is constant, but rather that it needs to be analyzed to understand our data. To clarify any confusion, we have revised the sentence (L119) as follows:

“The small splitting of 2.26 MHz in Fig. 5(a) arises from pre-existing non-zero strain parallel to the NV axis within the diamond crystal, which can vary from sample to sample”

  1. L208 “The lock-in signal in Fig. 5(c) is essentially the same as the data in Fig. 5(b), but we use the lock-in data due to its enhanced SNR, as high-frequency noise is filtered out by a low-pass filter in the lock-in amplifier”

– it is unclear what “lock-in” the authors are talking about, if above (L73) they stated that the oscillator frequency is fixed.

Authors: To avoid any confusion, we have revised the sentences as follows:

“The lock-in plot in Fig. 5(c) is obtained by recording the demodulated PL signal in response to an oscillating microwave field at a fixed frequency and repeating this process over the entire microwave frequency window. The lock-in data is essentially the same as that in Fig. 5(b), but we use the lock-in data for its enhanced SNR, as high-frequency noise is filtered out by the low-pass filter in the lock-in amplifier.”

  1. L215 “where 2.26 MHz the intrinsic splitting due to the crystal strain”

– see remark 5.

Authors: We have added the description “the splitting is not constant” in the sentence (L199).

  1. L218 “represent the maximum slope and the standard deviation of the lock-in signal”

– what is the nature of the noise that determines the value of the standard deviation? The authors do not mention this.

Authors: The standard deviation is an experimentally obtained value from the measured lock-in signal. Potential noise sources include fluctuations in the magnetic field and temperature, spurious coupling between microwave fields and photodetectors, etc.

  1. L234 “Although further improvements could be possible by increasing microwave power, we observed a sharp increase in sensitivity when the microwave power exceeded ~ 8 mW, which we suspect arises from the generation of eddy current noise in the PD due to the intense microwave field”

– does “increase in sensitivity” here mean deterioration or improvement?

Authors: We thank the reviewer for the misleading sentence. We have revised it as follows:

“Although further improvements could be possible by increasing microwave power, we observed a significant degradation in sensitivity when the microwave power exceeded ~ 8 mW, which we suspect arises from the generation of eddy current noise in the PD due to the intense microwave field”

  1. L311 “The fixed frequency measurement is commonly used in diamond magnetometry as it reduces total sensing time compared to the entire ESR spectrum throughout an wide frequency range”

– see note 4.

Authors: We have explained this in our response to note 4.

  1. L317 “The frequency point indicated as (1) is where the lock-in signal reaches its maximum. This frequency has been commonly employed due to the largest lock-in signal”

– This statement is absolutely incorrect, since this point is characterized by a zero derivative of the lock-in signal, and, accordingly, by the lack of sensitivity of this signal to changes in the magnetic field.

Authors: Frequency point (1) corresponds to the point where the slope of the ODMR signal is largest with respect to changes in the magnetic field (see Fig. 5(a)). Although the lock-in signal itself is flat, this point is commonly used when the magnetic field is large enough to alter the signal, and the maximum lock-in signal is advantageous in such cases. On the other hand, frequency point (2) offers better sensitivity for relatively small magnetic fields. These differences are discussed in Fig. 7(b) and the corresponding sections in the main text.

  1. L348 “In contrast, the region II (80 − 150 µT) shows the most pronounced and linear variation of the lock-in signal when measured at (1)”

– This happens precisely because the frequency, which in the zero field corresponded to point (1), ends up in a linear section as the field increases.

Authors: As mentioned in the previous response, each of the two frequency points (1) and (2) offers better performance within specific magnetic field ranges. We suggest selecting the appropriate frequency point based on the magnetic field. If the field is unknown or spans a wider range, our analysis method, which incorporates both linear and non-linear responses, will be useful for reconstructing the original magnetic profile.

  1. L351 “Figure 7(b) suggests that the optimal fixed frequency point should be adjusted according to the magnitude of BNV”

– this statement seems obvious, as well as all the results in this section.

Authors: As illustrated above, what may seem obvious has often been overlooked in previous research. Furthermore, we argue that these considerations should be included when using miniaturized diamond sensors to image areas with a magnetic field over a wide range.

  1. L392 “Since there are four different NV crystal axes and their ESR resonances overlap at non-zero or very weak 393 magnetic field, we calculated BNV for all the NV axes”

– the authors do not mention anywhere how they measured the position of the crystal axes relative to the coordinate system used.

Authors: To identify the crystal axes, we performed a separate measurement by applying an external magnetic field using a permanent magnet with known directions relative to the diamond sensor. We thank the reviewer for the suggestion, and we have revised the sentences as follows:

“Note that we identified the crystal axes beforehand in a separate measurement by applying an external magnetic field using a permanent magnet with known directions relative to the diamond crystal in our sensor configuration.”

  1. L433 “We propose that vector magnetometry based on NV centers is a powerful method for identifying unknown tilt angles and significantly reducing the errors”

– it is great that the authors fulfilled the reviewer’s request and added a numerical calculation, but they did not do the most important thing – namely, they did not show how a vector magnetometer scheme that works in the earth’s field can be implemented.

Authors: We proposed the toy diorama measurement to emulate real-world scenarios and highlighted some of potential issues and solutions. We agree that our experiment and simulation do not address all the challenges associated with real-world applications, such as differentiating signals from targets and the background Earth's field. Identifying various magnetic sources, such as Earth's field and man-made anomalies, requires prior knowledge of magnetic images taken before and after the targets are introduced. We think that this is an important topic for magnetic detection of hidden objects and presents similar challenges to those encountered with other magnetometers. Incorporating these topics into the vector magnetometry simulation is beyond the scope of our manuscript and could be the focus of more dedicated research in the future.

Overall, I don't think the article has been improved in any way by the revisions, and I still think it would be more suited to a popular journal rather than Sensors.

Authors: We believe that our work successfully demonstrates two-dimensional imaging with a miniaturized diamond sensor and provides detailed image analysis methods that are important for future research in real-world applications. Through multiple iterations, we have made every effort to address the questions and revise the manuscript according to the Reviewer’s suggestions. We do not believe that improving sensitivity or demonstrating performance in a real-world environment should be the sole focus of the journal. We respectfully ask the Reviewer to reconsider our manuscript.

Reviewer 2 Report (New Reviewer)

Comments and Suggestions for Authors

In this paper, authors used miniaturized NV magnetic sensors combined with a two- dimensional scanning stage to image concealed magnets in a toy diorama with millimeter- scale resolution.  Authors performed magnetic simulations to analyze the evolution of the lock-in data as a function of the magnetic field and found  good agreement between the measurements and simulations. Authors also  propose a method that uses vector magnetometry to compensate for the tilt angles of the  target, enabling accurate localization of its position. 

The article presents an interesting application of magnetic sensors based on diamonds with NV color centers. The article reads well and can be published after minor corrections.

Figure 2 () ODRM  -> ODMR

line 141 "... N ~" - Does "N" stand for nitrogen or for the number of color centers NV?

The authors provide the magnetic sensitivity, but could they provide the spatial resolution of the magnetometer?

6000 seconds of measurement is a very long time, do the authors have any idea how to shorten this time?

Author Response

Reviwer#2

Comments and Suggestions for Authors

In this paper, authors used miniaturized NV magnetic sensors combined with a two- dimensional scanning stage to image concealed magnets in a toy diorama with millimeter- scale resolution.  Authors performed magnetic simulations to analyze the evolution of the lock-in data as a function of the magnetic field and found  good agreement between the measurements and simulations. Authors also  propose a method that uses vector magnetometry to compensate for the tilt angles of the  target, enabling accurate localization of its position.

The article presents an interesting application of magnetic sensors based on diamonds with NV color centers. The article reads well and can be published after minor corrections.

Authors: We thank the Reviewer for their positive consideration.

1. Figure 2 () ODRM -> ODMR

Authors: We thank the Reviewer for pointing out the error. We have corrected it to “ODMR”.

2. line 141 "... N ~" - Does "N" stand for nitrogen or for the number of color centers NV?

Authors: In the revised version, we have corrected the sentence as follows:

“…, resulting in an NV concentration of N  ”

3. The authors provide the magnetic sensitivity, but could they provide the spatial resolution of the magnetometer?

Authors: Given that the sensor head measures 3 mm × 3 mm × 0.3 mm and the minimum distance between the sensor and the target is less than 1 cm, our setup theoretically offers a spatial resolution on the order of millimeters. However, in this study, the toy diorama is positioned approximately 10 cm from the sensor, meaning the spatial resolution described above is not fully utilized in this imaging demonstration. Note that the diorama measurement was conducted as a proof-of-principle to showcase the potential of locating hidden objects.

4. 6000 seconds of measurement is a very long time, do the authors have any idea how to shorten this time?

Authors: A total of 6,000 seconds were required to complete 1,200 steps over an imaging area of 24.38 cm × 18.75 cm. Each step took 5 seconds, with 1 second for measurement, 2 seconds for position adjustment, and 2 seconds for a pause before measurement. Enhancing sensitivity to reduce measurement time, along with improving the stability of the scanning setup, would help further decrease the overall time.

Reviewer 3 Report (New Reviewer)

Comments and Suggestions for Authors

The article demonstrates an example of scanning magnetic sensor based on ODMR effect in an ensemble of NV centers in diamond. The ODMR method itself is not new for magnetic sensing, so article concentrates on details of realization of sensor for relatively large (millimeter scale resolution) scanning/imaging, processing and interpretation of sensor data, and obtaining “vector” (directional) information about magnetic field using relatively simple sensor. The article is generally well written, provides a lot of details about experiment and data processing, I found no obvious issues with the article, so it clearly deserves publication.

Author Response

Reviwer#3

Comments and Suggestions for Authors

The article demonstrates an example of scanning magnetic sensor based on ODMR effect in an ensemble of NV centers in diamond. The ODMR method itself is not new for magnetic sensing, so article concentrates on details of realization of sensor for relatively large (millimeter scale resolution) scanning/imaging, processing and interpretation of sensor data, and obtaining “vector” (directional) information about magnetic field using relatively simple sensor. The article is generally well written, provides a lot of details about experiment and data processing, I found no obvious issues with the article, so it clearly deserves publication.

Authors: We thank the Reviewer for their positive feedback.

Round 2

Reviewer 1 Report (Previous Reviewer 3)

Comments and Suggestions for Authors

I am glad that the authors have changed the title and more clearly formulated the thesis that they do not claim to use their method to search for real mines on real battlefields.

And although I do not agree with all of their theses - for example, I do not understand why these mines are mentioned in the manuscript at all, if the proposed method uses a “Miniaturized Magnetometer” and is not scalable in principle - I am ready to stop discussing this topic.

I am also glad that to see the authors have made corrections to the text in accordance with some of my comments.

However, I cannot agree with some of the provisions in their responses to my comments, namely:

4. Authors: “In NV measurements, a full ESR spectrum provides a comprehensive understanding of the magnetic field, including both magnitude and direction” 

- scanning the entire spectrum is never used in real field devices, since this procedure is too long and does not provide the necessary signal-to-noise ratio. To obtain complete information about the magnitude and direction of the magnetic field, a known bias field is applied to the crystal, due to which the spectrum lines are resolved, four resonances corresponding to four orientations of the NV center axes are selected in the spectrum, and the frequencies of the four  generators are locked to them using the lock-in method.

6. Authors: “The lock-in plot in Fig. 5(c) is obtained by recording the demodulated PL signal in response to an oscillating microwave field at a fixed frequency;

Authors also write (L182): “ The PD signal is fed into a lock-in amplifier … and is demodulated at 5 MHz, which is concurrently employed to the DSRR to modulate the microwave fields” 

Thus, the expression “at a fixed frequency” (used 11 times in the text) misleads the reader, since it is unclear whether it refers to the microwave frequency or the modulation frequency. Further, if it refers to the microwave frequency, then we can only talk about a fixed carrier, but not about the frequency itself.

8. Authors: “Potential noise sources include fluctuations in the magnetic field and temperature, spurious coupling between microwave fields and photodetectors, etc” 

– Authors should know that the ultimate sensitivity of any optical quantum sensor is determined not by technical noise, which in principle can be eliminated, but by fundamental quantum noise – projection quantum noise and photon shot noise. Shot noise dominates in NV sensors, its value is easily calculated based on the intensity of the photocurrent. The shot noise sensitivity estimate is usually given along with the estimate of the actual sensitivity achieved. Comparison of these values gives an idea of the level of technical implementation of the sensor and the possibilities for its improvement.

11. Authors: “Frequency point (1) corresponds to the point where the slope of the ODMR signal is largest with respect to changes in the magnetic field (see Fig. 5(a))” 

– perhaps the authors mean that the slope of the “raw” (i.e. taken before lock-in amplification)  ODMR fluorescence signal is maximum at this point? If so, then in the text they should clearly indicate which type of signal (raw or processed by lock-in amplification) is used.

=======

Some comments on the new text of the manuscript:

1. L217-227, L343, L391, etc.: commas should be moved from the beginning of the line to the previous line (between the formula and its number).

2. L339: “We modeled the ODMR spectrum as a linear combination of two Lorentzian functions, including the intrinsic transverse splitting, and obtained the lock-in signal, Lsignal, as the derivative of the ODMR spectrum with respect to frequency 𝑓”

– this approximation is valid only in the limit of small modulation amplitude, and such an amplitude, as is known, does not provide optimal sensitivity.

3. L182: The PD signal is fed into a lock-in amplifier (MFLI, Zurich Instruments) and is demodulated at 5 MHz, which is concurrently employed to the DSRR to modulate the microwave fields

- why was such a high modulation frequency chosen? It should be taken into account that in addition to the time T2, which determines the ODMR line width of about 1-3 MHz, there is also the time T1, which is usually about 1-2 orders of magnitude greater than T2. Using a modulation frequency much greater than 1/T1 can lead to a significant loss of signal.

I would like to ask the authors, when making changes to the manuscript, to avoid, if possible, further growth of its volume.

Author Response

Reviewer#1

Comments and Suggestions for Authors

I am glad that the authors have changed the title and more clearly formulated the thesis that they do not claim to use their method to search for real mines on real battlefields.

And although I do not agree with all of their theses - for example, I do not understand why these mines are mentioned in the manuscript at all, if the proposed method uses a “Miniaturized Magnetometer” and is not scalable in principle - I am ready to stop discussing this topic.

I am also glad that to see the authors have made corrections to the text in accordance with some of my comments.

However, I cannot agree with some of the provisions in their responses to my comments, namely:

  1. Authors: “In NV measurements, a full ESR spectrum provides a comprehensive understanding of the magnetic field, including both magnitude and direction”

- scanning the entire spectrum is never used in real field devices, since this procedure is too long and does not provide the necessary signal-to-noise ratio. To obtain complete information about the magnitude and direction of the magnetic field, a known bias field is applied to the crystal, due to which the spectrum lines are resolved, four resonances corresponding to four orientations of the NV center axes are selected in the spectrum, and the frequencies of the four generators are locked to them using the lock-in method.

Authors: We understand the concern raised by the Reviewer. The original sentences were intended to assist those familiar with the full ESR spectrum typically used in NV experiments, while highlighting that sensor applications rely on measurements at a fixed point. To avoid any confusion, we have revised the sentences as follows:

L74 “In the measurements, we monitor changes in the PL signals at a constant carrier frequency, near the NV electron spin resonance (ESR).”

Note that, as mentioned in our response to note 6, we have defined the carrier frequency as  and used this notion consistently throughout of the manuscript.

  1. Authors: “The lock-in plot in Fig. 5(c) is obtained by recording the demodulated PL signal in response to an oscillating microwave field at a fixed frequency;

Authors also write (L182): “ The PD signal is fed into a lock-in amplifier … and is demodulated at 5 MHz, which is concurrently employed to the DSRR to modulate the microwave fields”

Thus, the expression “at a fixed frequency” (used 11 times in the text) misleads the reader, since it is unclear whether it refers to the microwave frequency or the modulation frequency. Further, if it refers to the microwave frequency, then we can only talk about a fixed carrier, but not about the frequency itself.

Authors: We thank the Reviewer for the note. We agree that the term “fixed frequency” could be misleading, so we have revised the manuscript to use  as the carrier frequency. Additionally, we have replaced  with . For example, the sentence in L210 has been updated as follows:

“The lock-in plot in Fig. 5(c) is obtained by recording the demodulated PL signal in response to an oscillating microwave field at , and repeating this process over the entire microwave frequency window by varying .”

  1. Authors: “Potential noise sources include fluctuations in the magnetic field and temperature, spurious coupling between microwave fields and photodetectors, etc”

– Authors should know that the ultimate sensitivity of any optical quantum sensor is determined not by technical noise, which in principle can be eliminated, but by fundamental quantum noise – projection quantum noise and photon shot noise. Shot noise dominates in NV sensors, its value is easily calculated based on the intensity of the photocurrent. The shot noise sensitivity estimate is usually given along with the estimate of the actual sensitivity achieved. Comparison of these values gives an idea of the level of technical implementation of the sensor and the possibilities for its improvement.

Authors: We agree with the Reviewer that understanding the ultimate noise level and characterizing the noise sources are crucial. We have confirmed that our measurements are not shot noise-limited, and we suspect that the additional noise primarily arises from spurious coupling between the microwave fields and the photodetectors (as mentioned in L241). However, since we have not yet implemented an active microwave filter design or fully diagnosed the noise sources, we reported the standard deviation of the measured lock-in signals rather than providing a detailed analysis of the noise. We acknowledge this and are actively working on eliminating these additional noise sources to further improve our sensor design.

  1. Authors: “Frequency point (1) corresponds to the point where the slope of the ODMR signal is largest with respect to changes in the magnetic field (see Fig. 5(a))”

– perhaps the authors mean that the slope of the “raw” (i.e. taken before lock-in amplification)  ODMR fluorescence signal is maximum at this point? If so, then in the text they should clearly indicate which type of signal (raw or processed by lock-in amplification) is used.

Authors: To avoid any confusion, we have revised the relevant sentence as follows:

L323 “This frequency has been commonly employed due to the maximum slope of the original ODMR spectrum and the largest lock-in signal.”

=======

Some comments on the new text of the manuscript:

  1. L217-227, L343, L391, etc.: commas should be moved from the beginning of the line to the previous line (between the formula and its number).

Authors: We have removed the commas.

  1. L339: “We modeled the ODMR spectrum as a linear combination of two Lorentzian functions, including the intrinsic transverse splitting, and obtained the lock-in signal, Lsignal, as the derivative of the ODMR spectrum with respect to frequency ?”

– this approximation is valid only in the limit of small modulation amplitude, and such an amplitude, as is known, does not provide optimal sensitivity.

Authors: As mentioned in our response to note 3 below, the modulation amplitude was determined by considering optical power, microwave driving and the hyperfine interaction for 14N. With the amplitude, we have not seen noticeable change in our signal. The model of two Lorentzian functions was validated at this modulation amplitude, as it yielded about the same results when smaller modulation amplitudes were used.

  1. L182: The PD signal is fed into a lock-in amplifier (MFLI, Zurich Instruments) and is demodulated at 5 MHz, which is concurrently employed to the DSRR to modulate the microwave fields

- why was such a high modulation frequency chosen? It should be taken into account that in addition to the time T2, which determines the ODMR line width of about 1-3 MHz, there is also the time T1, which is usually about 1-2 orders of magnitude greater than T2. Using a modulation frequency much greater than 1/T1 can lead to a significant loss of signal.

Authors: Generally speaking, the CW ODMR linewidth, , is limited by the coherence time , but it can also be broadened by the driving optical and microwave fields [Ref.1].

Based on our optical power, microwave driving, and the hyperfine interaction for 14N (), we determined the optimal modulation amplitude to be roughly [Ref.2]. We have done testes with slightly varying the values around 5 MHz and did not observed any noticeable loss in our signal. Note that modulation amplitudes of 1 – 5 MHz have been used in other sensing experiments based on diamond NV centers[Refs 3-5].

References for note 3:

  1. A Dreau, M Lesik, L Rondin, P Spinicelli, O Arcizet, J. F. Roch, and V Jacques, “Avoiding power broadening in optically detected magnetic resonance of single NV defects for enhanced dc magnetic eld sensitivity” Physical Review B 84, 195204 (2011)
  2. Alec A. Jenkins, “Probing condensed matter order with nitrogen-vacancy center scanning magnetometry” PhD Thesis, University of California Santa Barbara (2019)
  3. Yang Li, Doudou Zheng, Zhenhua Liu, Hui Wang, Yankang Liu, Chenyu Hou, Hao Guo, Zhonghao Li, Yashuhiro Sugawara, Jun Tang, Zongmin Ma, and Jun Liu, “Noise Suppression of Nitrogen-Vacancy Magnetometer in Lock-In Detection Method by Using Common Mode Rejection” Micromachines 14, 1823 (2023)
  4. Rolf Simon Schoenfeld and Wolfgang Harneit, “Real Time Magnetic Field Sensing and Imaging Using a Single Spin in Diamond” Physical Review Letters 106, 030802 (201)
  5. Tianning Wang, Zhenhua Liu, Yankang Liu, Bo Wang, Yuanyuan Shen, and Li Qin, “High-Dynamic-Range Integrated NV Magnetometers” Micromachines15, 662 (2024)

I would like to ask the authors, when making changes to the manuscript, to avoid, if possible, further growth of its volume.

Authors: We thank the Reviewer for their valuable insights and suggestions. With the feedback, our manuscript has become clearer and more robust.

This manuscript is a resubmission of an earlier submission. The following is a list of the peer review reports and author responses from that submission.

Round 1

Reviewer 1 Report

Comments and Suggestions for Authors

This paper should present the details of the prepare process of diamond nitrogen-vacancy (NV) centers, which is very importeant to magnetometer with a suitable NV array.

The authors should provide the details of process method of diamond NV, which is very important for sensor manufacturing. Comments on the Quality of English Language

The quality of English in this paper shoud be inmproved. 

Author Response

Reviwer#1

Comments and Suggestions for Authors

This paper should present the details of the prepare process of diamond nitrogen-vacancy (NV) centers, which is very important to magnetometer with a suitable NV array.

The authors should provide the details of process method of diamond NV, which is very important for sensor manufacturing.

Authors: We thank the Reviewer for their valuable feedback. We have provided more detailed information about the diamond NV process in Section 2.2., as follows:

“We used a commercial diamond plate from Element Six; single crystal type 1b with the dimension of 3 mm  3 mm  0.3 mm and a <100> crystal direction, and a nitrogen concentration of < 200 ppm. The diamond was irradiated with 1.8  1015 electrons at 2 MeV and subsequently annealed at 900°C for 2 hours, resulting in an NV concentration of N  .”

The quality of English in this paper should be improved. 

Authors: We have carefully reviewed the manuscript and made revisions to the English to improve its clarity.

Reviewer 2 Report

Comments and Suggestions for Authors

  The authors reported an experimental setup for magnetic field imaging and demonstrated the experimental results. This device used the negatively charged nitrogen vacancy color center ensemble in diamond as a magnetic sensor, which can image targets on the order of ten centimeters in size. The authors demonstrated the construction and the working principle based on phase-locked measurement, analyzed the relationship between microwave power and sensitivity, and imaged magnets hidden under toys. The authors also analyzed the dynamic range of magnetic imaging and achieved magnetic imaging of a magnet. The experimental process of the entire article is rigorous and there are no logical issues.

  As a demonstration of macroscopic imaging based on NV color centers, the imaging capability of small magnets was demonstrated. As the authors described in the manuscript, this technique has not been reported yet. However, my concern is that although the authors have conducted a thorough exploration of the system, the poor sensitivity demonstrated in this work will greatly affect the innovation and impact of this article. The sensitivity of ~0.6 microTesla in the article is even worse than some single NV center. With the increase of microwave power, this sensitivity value cannot be improved to a higher level. The reason for this result is that the author did not properly decouple the microwave from the photodetector, resulting in strong noise caused by the coupling of microwave pulses into the detection circuit.

  In practice where this technique is potentially used, the detection target is a ferromagnetic substance hidden inside the sample, whose magnetism is several orders of magnitude weaker than that of a magnet. Due to poor sensitivity and imaging speed, only some magnets with strong magnetism can be measured as experimental demonstrations, which cannot be used for practical applications. This is another factor that limits the influence of this article.

  In summary, I don’t recommend the paper to be published at the current form. I encourage the author to further optimize the setup, such as decoupling the data acquisition of the detector and microwave pulses in the time domain to avoid the influence of noise, or use pulse ODMR procedure to aboid this noise, in order to further enhance the practicality of the detector and meet the scope of the journal.

Author Response

Reviewer#2

Comments and Suggestions for Authors

The authors reported an experimental setup for magnetic field imaging and demonstrated the experimental results. This device used the negatively charged nitrogen vacancy color center ensemble in diamond as a magnetic sensor, which can image targets on the order of ten centimeters in size. The authors demonstrated the construction and the working principle based on phase-locked measurement, analyzed the relationship between microwave power and sensitivity, and imaged magnets hidden under toys. The authors also analyzed the dynamic range of magnetic imaging and achieved magnetic imaging of a magnet. The experimental process of the entire article is rigorous and there are no logical issues.

As a demonstration of macroscopic imaging based on NV color centers, the imaging capability of small magnets was demonstrated. As the authors described in the manuscript, this technique has not been reported yet. However, my concern is that although the authors have conducted a thorough exploration of the system, the poor sensitivity demonstrated in this work will greatly affect the innovation and impact of this article. The sensitivity of ~0.6 microTesla in the article is even worse than some single NV center. With the increase of microwave power, this sensitivity value cannot be improved to a higher level. The reason for this result is that the author did not properly decouple the microwave from the photodetector, resulting in strong noise caused by the coupling of microwave pulses into the detection circuit.

Authors: We thank the Reviewer for their valuable feedback. As the Reviewer noted, the sensitivity demonstrated in this paper is comparable to that observed in early miniaturized sensors, though not as high as the most recent advancements. However, improving sensitivity is not the primary focus of this study, nor did we claim it to be in our paper. Instead, our main objective was to use the sensor for imaging objects over large areas (> cm²) with millimeter-scale resolution, while developing more comprehensive analysis methods to interpret magnetic images, particularly when the field magnitude varies over a wide range, leading to image distortion. While we acknowledge that sensitivity is an important subject, we believe that not all papers need to focus on this aspect, especially when other critical challenges, like imaging analysis, are being addressed.

We agree with the Reviewer that microwave fields can introduce spurious noise in the adjacent photodetector, which we indeed observed. Addressing the decoupling or suppression of this noise will be the focus of future investigations. We have been developing newer sensor versions where the photodetector is externally coupled via fiber. However, as stated earlier, these improvements will be the subject of another publications. The primary focus of this manuscript remains on addressing the challenges of using miniaturized sensors for imaging.

In practice where this technique is potentially used, the detection target is a ferromagnetic substance hidden inside the sample, whose magnetism is several orders of magnitude weaker than that of a magnet. Due to poor sensitivity and imaging speed, only some magnets with strong magnetism can be measured as experimental demonstrations, which cannot be used for practical applications. This is another factor that limits the influence of this article.

Authors: We agree with the Reviewer that improved sensitivity is crucial for the practical applications of miniaturized sensors for the target detection. Previous studies, such as Mu, Y et al. (Remote Sensing, 2020, 12, 452), suggest that the magnetic fields from buried mines, measured a few meters above the ground, are on the order of 1 nT – 100 nT. This would require a significant improvement in the sensitivity of the current sensors presented in this paper. However, as mentioned earlier, sensitivity is not the primary focus of this study. We anticipate that the sensitivity will reach the levels required for practical applications, based on recent advancements in miniaturized diamond sensors and our newer sensor versions, even though these improvements are not discussed in this paper (still in progress). In this study, we used the diorama sample to simulate relevant scenarios, with a focus on image analysis rather than addressing the sensitivity for practical applications at this stage.

We have added the following sentences in Section 2.4.

“Previous studies suggest that the magnetic fields from buried mines, measured a few me-ters above the ground, fall within the range of 1 nT to 100 nT [36-40]. Achieving this level of sensitivity would require several orders of magnitude improvement over the current sensors presented in this paper. However, we anticipate that sensitivity can reach the necessary levels for practical applications, given recent advancements in miniaturized diamond sensors [10-14]. In this study, we used a toy diorama sample to simulate relevant battlefield scenarios, focusing on image analysis rather than addressing sensitivity for practical applications at this stage.”

In summary, I don’t recommend the paper to be published at the current form. I encourage the author to further optimize the setup, such as decoupling the data acquisition of the detector and microwave pulses in the time domain to avoid the influence of noise, or use pulse ODMR procedure to aboid this noise, in order to further enhance the practicality of the detector and meet the scope of the journal.

Authors: We thank the Reviewer for the suggestion. However, as mentioned several times, the primary aim of this paper is to address the imaging challenges and provide a more comprehensive analysis. Improving the setup and addressing the issues related to the current sensor design are topics for future publications, and we are actively working on them.

Reviewer 3 Report

Comments and Suggestions for Authors

The manuscript “Millimeter-Resolution Scanning Miniaturized Magnetometer Based On Diamond Quantum Sensor” is devoted to the experimental proof of the applicability of magnetic field sensors on NV centers for magnetic scanning of surfaces. The article is well structured, well written and provided with extensive illustrative material, as a result of which it is read with interest.

However, I don't think it should be published in the Sensors journal.

First of all, I should note that the stated goal of the work in the abstract (“By scanning over a toy diorama with hidden magnets, we simulate scenarios such as remote detection of landmines on a battlefield or locating concealed objects at a construction site”) raises some doubts - how do the authors propose to scale the size of their search setup when moving from a “toy diorama” to real “battlefields”?

Comments and remarks:

1. P3 Fig2b does not show singlet states and intercombination transitions, which play a decisive role in creating the population difference.

2. P3 Fig 2c (lower spectrum) – it is unclear why the spectral peaks are not split into triplets due to hyperfine interaction. It should also be noted that there are no scales along the axes, which is especially important for the horizontal axis.

3. P4 “The resonator’s return loss measurement, S11, depicted in Fig. 4(a), indicates a resonance at 2.87 GHz with a quality factor Q of 395”

 – from which it follows that the resonator bandwidth is only 7 MHz. Is this sufficient, given the spectral line width, gyromagnetic ratio, and modulation frequency of 5 MHz?

4. The design of the sensor in Fig.3 does not seem optimal to me, since the sensitive element (diamond) is placed closest to the wall on which the microwave connector is located - which does not allow, if necessary, to bring the sensitive element as close as possible to the object (it should be taken into account that the magnetic dipole field decreases as the cube of the distance!)

5. P5 “The quality factor Q is reduced to 160 due to the insertion of the copper”

- this, of course, partially answers remark #3, but how reliable is this method of frequency shifting, and does it lead to a shift in the spatial distribution of the microwave field?

6. P5 “The sensitivity of the magnetic field, 𝜂B, from the lock-in data can be calculated using the equation”

(the equation is not numbered). The given formula is valid only on the linear section of the frequency dependence on the magnetic field. In fields less than a few Gs, the sensitivity is significantly reduced due to transverse splitting, in this case amounting to 2.26 MHz.

7. P5 “showing the best sensitivity of 𝜂B= 628 ± 3 𝑛𝑇/√Hz”

– this value is three orders of magnitude worse than the values ​​usually obtained on bulk samples. In fact, the achieved sensitivity does not exceed the sensitivity of the Hall sensor in a mobile phone.

It also contradicts the statements above: “sensitivity of approximately 1 μT/√Hz for a single NV center”, “Ensemble NV centers are commonly utilized in compact magnetic sensors, providing enhanced sensitivity by a factor of √𝑁 for shot noise-limited measurements, with N denoting the number of NV centers” (P3) and “an effective number of NV centers as ~ 3×10^13” (P4).

If we divide the value of 1 μT/√Hz by √𝑁, where N = 3×10^13, we get the expected sensitivity of the order of 0.2 pT/√Hz. Is the discrepancy with the sensitivity obtained by the authors 𝜂B= 628 ± 3 𝑛𝑇/√Hz explained by the operation in low fields in the absence of a bias field (see the previous remark), or by something else?

8. P7 “To our knowledge, the NV-based magnetometer has not yet been demonstrated for this purpose; however, its compact size, high sensitivity, and ability for vector magnetometry indicate potential applications in military and industrial sectors”

 – As follows from the text, the scheme used by the authors does not have any “abilities for vector magnetometry”.

9. P10 Fig 9a “A schematic of relative angle between the NV centers in a miniaturized sensor 357 and a magnet”

– in this context it is necessary to specify what exactly is tilted – the sensor or the magnet. The shift in the field distribution maximum is caused by the tilt of the magnet, but not the sensor.

10. P10 Fig 9b “Calculated magnetic field from the magnet at z = 10”

– no units of measurement are given. 

11. P11 The authors demonstrate the shift in the measurement results when the magnetic field source is tilted, but do not offer any methods for compensating for it – which could be easily done using the ability of NV centers to take vector field measurements.

To summarize the above:

·         the authors use a sensor design that is far from optimal and is characterized by extremely low sensitivity - worse than the Hall sensor in a mobile phone, but at the same time lacking the ability to determine the direction of the magnetic field vector;

·         the authors do not offer any methods for increasing the sensitivity to at least nanotesla units;

·         the authors do not use the inherent ability of NV centers to measure vector fields, resulting in an uncontrolled shift in the result.

·         The ability of the setup assembled by the authors to detect magnetic fields at a level of 50 μT (!!!) is also not very impressive.

·         It is absolutely unclear how the scheme used by the authors can be scaled up to the level of real objects (landmines and battlefields, for which the authors propose to use it).

The most important thing is that in the entire article I was unable to see a single truly original scientific or technical solution.

Conclusion: the manuscript, although it is of literary and educational interest, is unlikely to correspond in its level to the Sensors journal, and should be redirected to a popular science or pedagogical journal.

Author Response

Reviewer#3

The manuscript “Millimeter-Resolution Scanning Miniaturized Magnetometer Based On Diamond Quantum Sensor” is devoted to the experimental proof of the applicability of magnetic field sensors on NV centers for magnetic scanning of surfaces. The article is well structured, well written and provided with extensive illustrative material, as a result of which it is read with interest.

However, I don't think it should be published in the Sensors journal.

First of all, I should note that the stated goal of the work in the abstract (“By scanning over a toy diorama with hidden magnets, we simulate scenarios such as remote detection of landmines on a battlefield or locating concealed objects at a construction site”) raises some doubts - how do the authors propose to scale the size of their search setup when moving from a “toy diorama” to real “battlefields”?

Authors:

Authors: We would like to first thank the Reviewer for their valuable feedback. We agree that improved sensitivity is crucial for the practical applications of miniaturized sensors in target detection on “battlefields”. Previous studies, such as Mu, Y et al. (Remote Sensing, 2020, 12, 452), suggest that the magnetic fields from buried mines, measured a few meters above the ground, are in the range of 1 nT – 100 nT. This would require a few orders of magnitude improvement in the sensitivity of the current sensors presented in this paper. However, the primary focus of this study is not sensitivity but rather imaging analysis. We anticipate that sensitivity will reach the required levels for practical applications, based on recent advancements in miniaturized diamond sensors and our newer sensor versions, though these improvements are not discussed in this paper (still in progress). In this study, we used a toy diorama sample to simulate relevant battlefield scenarios, with a focus on image analysis rather than addressing sensitivity for practical applications at this stage.

We have added the following sentences in Section 2.4.

“Previous studies suggest that the magnetic fields from buried mines, measured a few me-ters above the ground, fall within the range of 1 nT to 100 nT [36-40]. Achieving this level of sensitivity would require several orders of magnitude improvement over the current sensors presented in this paper. However, we anticipate that sensitivity can reach the necessary levels for practical applications, given recent advancements in miniaturized diamond sensors [10-14]. In this study, we used a toy diorama sample to simulate relevant battlefield scenarios, focusing on image analysis rather than addressing sensitivity for practical applications at this stage.”

Comments and remarks:

  1. P3 Fig2b does not show singlet states and intercombination transitions, which play a decisive role in creating the population difference.

Authors: We thank the Reviewer for the suggestion and agree that the intersystem transitions via the singlet states are important. We have incorporated this suggestion and revised the figure accordingly.

Revised Fig. 2 (b)

  1. P3 Fig 2c (lower spectrum) – it is unclear why the spectral peaks are not split into triplets due to hyperfine interaction. It should also be noted that there are no scales along the axes, which is especially important for the horizontal axis.

Authors: The ODMR spectra in Fig. 2C are schematic representations, not actual data, intended to illustrate the differences between the ODMR measured for a single NV center and an ensemble of NV centers. The schematics illustrate only the Zeeman splitting due to an external magnetic field and do not include the hyperfine interaction. For the x-axis, we have only included the center frequency of 2870 MHz, as the spectrum is not based on real data. To avoid any confusion, we have updated the figure caption to include the clarification: 'Schematics' of the ODMR spectra...'.

  1. P4 “The resonator’s return loss measurement, S11, depicted in Fig. 4(a), indicates a resonance at 2.87 GHz with a quality factor Q of 395”

 – from which it follows that the resonator bandwidth is only 7 MHz. Is this sufficient, given the spectral line width, gyromagnetic ratio, and modulation frequency of 5 MHz?

Authors: The spectral linewidth of the NV resonances is approximately 12 MHz, which is larger than the resonator bandwidth of 7 MHz and can affect the lock-in measurement of the full ESR resonances. However, as discussed in the manuscript and shown in Fig. 4(b), we added a copper block to match the frequencies of the DSRR resonator and the NV resonance. This addition reduces the quality factor (Q) to 160, corresponding to a bandwidth of approximately 18 MHz, which enables us to perform the lock-in measurement over the entire ESR spectrum.

We agree that a high Q factor of the resonator is beneficial for generating larger microwave fields, but it is effective only for narrow bandwidths, making it suitable for monitoring small changes in the magnetic field. For measuring a broader range of signals, alternative microwave circuitries, such as waveguides, would be more appropriate. However, as demonstrated above, the Q factor of 160 is sufficiently broad for our current experiment.

  1. The design of the sensor in Fig.3 does not seem optimal to me, since the sensitive element (diamond) is placed closest to the wall on which the microwave connector is located - which does not allow, if necessary, to bring the sensitive element as close as possible to the object (it should be taken into account that the magnetic dipole field decreases as the cube of the distance!)

Authors: We agree with the Reviewer that the thickness of the DSRR and the housing set the minimum distance between the diamond and the sample. If the goal is to use the sensors to detect proximity to samples, a better design would be required to minimize this distance. Our ongoing efforts are focused on improving both the sensitivity and the overall design. However, as previously mentioned, the primary aim of this paper is to focus on magnetic field imaging with the miniaturized sensor, specifically through imaging analysis, particularly over a wide range of magnetic fields from the samples.

  1. P5 “The quality factor Q is reduced to 160 due to the insertion of the copper”

- this, of course, partially answers remark #3, but how reliable is this method of frequency shifting, and does it lead to a shift in the spatial distribution of the microwave field?

Authors: Although the DSRR is designed to have a center frequency aligned to 2.87 GHz, the frequency may shift after fabrication. To tune the frequency and match it to the NV resonance, we inserted a copper plate near the resonator, following the method used in Ref. 45: Stürner et al., Adv. Quantum Tech. 2021, 4, 2000111. We manually adjusted the copper’s position until the frequencies were matched.

The insertion of the copper plate does affect the DSRR performance, and we observed a reduction in the Q factor, which in turn reduces the magnitude of the microwave fields. While the overall profile of the microwave fields can change, our simulations show that the uniformity of the fields within the NV center detection volume inside the diamond is not significantly affected (see the image below). Therefore, despite the reduction in the overall microwave field magnitude, the spatial distribution remains largely unaffected in our measurements.

Moreover, during the revision, we noticed that the location of the copper plate in the inset of Fig. 4(a) was incorrect and have corrected it in the revised figure.

Figure. Microwave field simulations before (left) and after (right) the insertion of a copper plate. From top to bottom, the cross-sectional views are shown along the xy, yz, and xz planes. For the comparisons, we used the same scale for the field intensity (dB).

  1. P5 “The sensitivity of the magnetic field, ?B, from the lock-in data can be calculated using the equation”

(the equation is not numbered). The given formula is valid only on the linear section of the frequency dependence on the magnetic field. In fields less than a few Gs, the sensitivity is significantly reduced due to transverse splitting, in this case amounting to 2.26 MHz.

Authors: We agree that the explanation was insufficient. Indeed, the sensitivity equation is only valid in the linear regime, where the magnetic field exceeds the transverse splitting caused by the intrinsic strain (2.26 MHz). Since the sensitivity depends on the relative magnitudes of the transverse splitting and the magnetic field, we have provided the sensitivity in the linear regime as an example. We have added a discussion on this point in Section 2.3., as follows:

“The Zeeman shift of the NV resonances can be written as:

 (1)

where  the intrinsic splitting due to the crystal strain. The sensitivity of the magnetic field, , from the lock-in data can be calculated using the equation:

 (2)

, where  and  represent the maximum slope and the standard deviation of the lock-in signal, respectively, and  is the interrogation time, i.e., 1 second[34,35]. If , Eq. 2 can be approximated as:

 (3)

, which assumes a linear dependence of the Zeeman on . On the other hand, if , Eq. 2 can be approximated as:

 (4)

, which introduces an additional factor of , meaning the sensitivity also depends on . Although we do not apply external magnetic fields to split the NV resonances beforehand in our current experiments, the linear regime of Eq. 3 can be achieved either by applying an external magnetic field in advance or by ensuring the magnetic field to be probed is larger than the intrinsic transverse splitting.

To illustrate the sensitivity of the sensors used in the study, we calculate the sensitivity using Eq. 3, assuming a linear dependence of the Zeeman shift on .”

  1. P5 “showing the best sensitivity of ?B= 628 ± 3 ??/√Hz”

– this value is three orders of magnitude worse than the values ​​usually obtained on bulk samples. In fact, the achieved sensitivity does not exceed the sensitivity of the Hall sensor in a mobile phone.

Authors: We agree that the sensitivity demonstrated in this paper is comparable to that observed in early miniaturized sensors, but not as high as the most recent measurements. One of the primary reasons for this is the close proximity between the microwave sources and photodetectors. There are several design approaches to improve the sources and detectors, or to separate them using fiber-coupled external detectors. In our paper, we did not claim that our sensor improves sensitivity compared to the most recent results. Instead, we emphasized addressing imaging challenges and providing a more comprehensive analysis, rather than introducing a new sensor design. Improving the setup and addressing the issues related to the current sensor design are topics for future publications, and we are actively working on them.

It also contradicts the statements above: “sensitivity of approximately 1 μT/√Hz for a single NV center”, “Ensemble NV centers are commonly utilized in compact magnetic sensors, providing enhanced sensitivity by a factor of √? for shot noise-limited measurements, with N denoting the number of NV centers” (P3) and “an effective number of NV centers as ~ 3×10^13” (P4).

If we divide the value of 1 μT/√Hz by √?, where N = 3×10^13, we get the expected sensitivity of the order of 0.2 pT/√Hz. Is the discrepancy with the sensitivity obtained by the authors ?B= 628 ± 3 ??/√Hz explained by the operation in low fields in the absence of a bias field (see the previous remark), or by something else?

Authors: The sentences illustrate the general properties of NV centers and suggest potentials of the sensor based on ensemble NV centers. However, to prevent any potential misunderstanding, we have added a sentence clarifying that the current sensitivity of our sensors is not optimal, and we also discuss potential improvements in sensor design.

The discrepancy between the simple estimation of 0.2 pT/√Hz and the measured 628 nT/√Hz can arise from several factors. In addition to the differences in coherence properties between single and ensemble NV centers, both optical and microwave power can influence the sensitivity through heating or spurious noise coupling. For instance, we found that eddy-current-induced noise coupling between the microwave circuits and the photodetectors, when in close proximity, hinders further improvement in our sensitivity.

It is worth noting that when we used the same diamond sample and similar optical and microwave power, but with a table-top experimental setup rather than a miniaturized one, we typically measured a few nT/√Hz. As miniaturizing and packing the components can degrade sensitivity, addressing this issue is crucial and remains a topic for future studies.

The following sentences have been added to Section 2.1.:

“Ensemble NV centers are commonly utilized in compact magnetic sensors, which, in principle, can provide enhanced sensitivity by a factor of √N for shot noise-limited measurements, with N denoting the number of NV centers[2,3]. However, in real experiments, differences in coherence properties between single and ensemble NV centers, as well as spurious coupling effects in miniaturized sensor designs, can affect the extent of sensitivity enhancement. Improving sensitivity and addressing the issues of miniaturization remain important subjects for future studies.”

  1. P7 “To our knowledge, the NV-based magnetometer has not yet been demonstrated for this purpose; however, its compact size, high sensitivity, and ability for vector magnetometry indicate potential applications in military and industrial sectors”

 – As follows from the text, the scheme used by the authors does not have any “abilities for vector magnetometry”.

Authors: The sentences also highlight the general potential of NV sensors for vector magnetometry. However, as we did not intentionally apply external magnetic fields in this paper, we did not utilize the four axes of the NV centers for vector magnetometry. In this paper, our aim was to use the NV centers without applying a magnetic field in advance, and use them to probe magnetic fields from targets. To incorporate vector magnetometry, we could add a permanent magnet or an active coil design along with the miniaturized sensors, but this would impact the overall size of the sensor and require the design of a new sensor. Moreover, we used a DSRR microwave resonator designed to have the center frequency matching to the NV’s zero-field splitting.

  1. P10 Fig 9a “A schematic of relative angle between the NV centers in a miniaturized sensor and a magnet”

– in this context it is necessary to specify what exactly is tilted – the sensor or the magnet. The shift in the field distribution maximum is caused by the tilt of the magnet, but not the sensor.

Authors: It refers to the tilt of the magnet relative to the sensor's bottom surface. We have added this clarification in the caption, as follows:

“Calculated magnetic field at z = 10 cm, , obtained from the tilted magnet relative to the sensor.”

  1. P10 Fig 9b “Calculated magnetic field from the magnet at z = 10”

– no units of measurement are given. 

Authors: We thank the Reviewer for identifying the error. We have added the unit ‘cm’ in the caption.

  1. P11 The authors demonstrate the shift in the measurement results when the magnetic field source is tilted, but do not offer any methods for compensating for it – which could be easily done using the ability of NV centers to take vector field measurements.

Authors: We agree that tilting can be compensated by rotating the sensor or, more actively, by utilizing the capabilities of vector magnetometry. As mentioned in Response 8, we did not employ vector magnetometry in this work, but instead focused on testing the imaging analysis associated with targets that may be tilted relative to the sensor in real applications. The Reviewer’s suggestion provides an interesting direction for future development.

To summarize the above:

  • the authors use a sensor design that is far from optimal and is characterized by extremely low sensitivity - worse than the Hall sensor in a mobile phone, but at the same time lacking the ability to determine the direction of the magnetic field vector;
  • the authors do not offer any methods for increasing the sensitivity to at least nanotesla units;
  • the authors do not use the inherent ability of NV centers to measure vector fields, resulting in an uncontrolled shift in the result.
  • The ability of the setup assembled by the authors to detect magnetic fields at a level of 50 μT (!!!) is also not very impressive.
  • It is absolutely unclear how the scheme used by the authors can be scaled up to the level of real objects (landmines and battlefields, for which the authors propose to use it).

The most important thing is that in the entire article I was unable to see a single truly original scientific or technical solution.

Conclusion: the manuscript, although it is of literary and educational interest, is unlikely to correspond in its level to the Sensors journal, and should be redirected to a popular science or pedagogical journal.

Round 2

Reviewer 2 Report

Comments and Suggestions for Authors

The authors have answered all the questions. I agree for publication.

Author Response

Comments and Suggestions for Authors

The authors have answered all the questions. I agree for publication.

Authors: We thank the Reviewer for the positive feedback.

Reviewer 3 Report

Comments and Suggestions for Authors

I am glad to see that the authors took the reviewer’s comments seriously and made a number of changes to the manuscript that significantly improved it. However, I believe that some of the comments are still relevant.

For example, the authors write: “In this study, we used a toy diorama sample to simulate relevant battlefield scenarios, with a focus on image analysis rather than addressing sensitivity for practical applications at this stage”.

This statement completely shifts the focus of the article and should be placed in the abstract rather than in section 2.4. In addition, given this statement, the title of the article “Millimeter-Resolution Scanning Miniaturized Magnetometer Based On Diamond Quantum Sensor” does not reflect its essence.

Further, the authors added the following fragment to the manuscript: “Previous studies suggest that the magnetic fields from buried mines, measured a few meters above the ground, fall within the range of 1 nT to 100 nT [36-40]. Achieving this level of sensitivity would require several orders of magnitude improvement over the current sensors presented in this paper”. The authors do not mention that when searching for real mines, the millimeter resolution of the magnetometer mentioned in the title of the article becomes completely meaningless.

Further, the authors write: “We agree that tilting can be compensated by rotating the sensor or, more actively, by utilizing the capabilities of vector magnetometry. As mentioned in Response 8, we did not employ vector magnetometry in this work, but instead focused on testing the imaging analysis associated with targets that may be tilted relative to the sensor in real applications”. It seems to me that even in the absence of the possibility of experimentally demonstrating the capabilities of vector magnetometry, the authors could have conducted a numerical simulation and shown the fundamental possibility of eliminating the error in determining the location.

The lack of new scientific or engineering solutions still seems to me to be the main shortcoming of the manuscript.

Author Response

Reviewer#3

Comments and Suggestions for Authors

I am glad to see that the authors took the reviewer’s comments seriously and made a number of changes to the manuscript that significantly improved it. However, I believe that some of the comments are still relevant.

For example, the authors write: “In this study, we used a toy diorama sample to simulate relevant battlefield scenarios, with a focus on image analysis rather than addressing sensitivity for practical applications at this stage”.

This statement completely shifts the focus of the article and should be placed in the abstract rather than in section 2.4. In addition, given this statement, the title of the article “Millimeter-Resolution Scanning Miniaturized Magnetometer Based On Diamond Quantum Sensor” does not reflect its essence.

Authors: We thank the Reviewer for the positive feedback. In response to the Reviewer’s suggestion, we have corrected the title to “Scanning Miniaturized Magnetometer Based On Diamond Quantum Sensors and Image Analysis for Target Localization”, and revised the abstract as follows:

“We have developed a miniaturized magnetic sensor based on diamond nitrogen-vacancy (NV) centers, combined with a two-dimensional scanning setup that enables imaging of magnetic samples with millimeter-scale resolution. Using the lock-in detection scheme, we tracked changes in the NV’s spin resonances induced by the magnetic field from target samples. As a proof-of-principle demonstration of magnetic imaging, we used a toy diorama with hidden magnets to simulate scenarios such as remote detection of landmines on a battlefield or locating concealed objects at a construction site, focusing on image analysis rather than addressing sensitivity for practical applications. The obtained magnetic images reveal that they can be influenced and distorted by the choice of frequency point used in the lock-in detection, as well as the magnitude of the sample’s magnetic field. Through magnetic simulations, we find good agreement between the measured and simulated images. Additionally, we propose a method based on NV vector magnetometry to compensate for the non-zero tilt angles of a target, enabling accurate localization of its position. This work introduces a novel imaging method using a scanning miniaturized magnetometer to detect hidden magnetic objects, with potential applications in military and industrial sectors.”

Further, the authors added the following fragment to the manuscript: “Previous studies suggest that the magnetic fields from buried mines, measured a few meters above the ground, fall within the range of 1 nT to 100 nT [36-40]. Achieving this level of sensitivity would require several orders of magnitude improvement over the current sensors presented in this paper”. The authors do not mention that when searching for real mines, the millimeter resolution of the magnetometer mentioned in the title of the article becomes completely meaningless.

Authors: We used the battle field scenario as an example to demonstrate the application of the scanning setup. In this context, we agree that millimeter resolution is not a key factor, which is why we have de-emphasized it by changing the title. However, it’s important to note that the setup can be applied to other samples where millimeter-resolution imaging is achievable in principle.

Further, the authors write: “We agree that tilting can be compensated by rotating the sensor or, more actively, by utilizing the capabilities of vector magnetometry. As mentioned in Response 8, we did not employ vector magnetometry in this work, but instead focused on testing the imaging analysis associated with targets that may be tilted relative to the sensor in real applications”. It seems to me that even in the absence of the possibility of experimentally demonstrating the capabilities of vector magnetometry, the authors could have conducted a numerical simulation and shown the fundamental possibility of eliminating the error in determining the location.

The lack of new scientific or engineering solutions still seems to me to be the main shortcoming of the manuscript.

Authors: We thank the Reviewer for the valuable suggestion. In response, we have conducted additional simulations to identify unknown tilt angles using NV vector magnetometry, significantly reducing the position errors of the target. This new work is now mentioned in the abstract, main text, and conclusion. Below is the revised content in the main text.